# Transformer Block Coupling and its Correlation with Generalization in LLMs

**Murdock Aubry**[*]  **Haoming Meng**[*]  **Anton Sugolov**[*]  **Vardan Papyan**
University of Toronto

## Abstract

Large Language Models (LLMs) have made significant strides in natural language processing, and a precise understanding of the internal mechanisms driving their success is essential. In this work, we analyze the trajectories of token embeddings as they pass through transformer blocks, linearizing the system along these trajectories through their Jacobian matrices. By examining the relationships between these block Jacobians, we uncover the phenomenon of **transformer block coupling** in a multitude of LLMs, characterized by the coupling of their top singular vectors across tokens and depth. Our findings reveal that coupling *positively correlates* with model performance, and that this relationship is stronger than with other hyperparameters such as parameter count, model depth, and embedding dimension. We further investigate how these properties emerge during training, observing a progressive development of coupling, increased linearity, and layer-wise exponential growth in token trajectories. Additionally, experiments with Vision Transformers (ViTs) corroborate the emergence of coupling and its relationship with generalization, reinforcing our findings in LLMs. Collectively, these insights offer a novel perspective on token interactions in transformers, opening new directions for studying their mechanisms as well as improving training and generalization.

## 1 Introduction

Transformers (Vaswani et al., 2017) can be represented as discrete, nonlinear, coupled dynamical systems, operating in high dimensions (Greff et al., 2016; Papyan et al., 2017; Haber & Ruthotto, 2017; Ee, 2017; Ebski et al., 2018; Chen et al., 2018; Bai et al., 2019; Rothauge et al., 2019; Gai & Zhang, 2021; Li & Papyan, 2023). Viewing the skip connections as enabling a discrete time step, we represent the hidden representations as dynamically evolving through the layers of the network. The term *nonlinear* refers to the nonlinear transformations introduced by activation functions, and *coupled* refers to the interdependent token trajectories that interact through the MLP and self-attention layers.

In our work, we investigate whether there are identifiable structural characteristics across 30+ pretrained LLMs, analyze their emergence with training, and examine their relationship with generalization performance. During inference, as token embeddings pass through the network, we analyze their layer-wise trajectories and linearize the effect of transformer blocks on the token embeddings across model depth to study the relationships between blocks. To this end, we compute the Jacobians of distinct connections between layers and tokens, perform singular value decompositions (SVDs), and compare the resulting singular vectors. This approach measures the degree of coupling between singular vectors to capture the operational coordination of blocks as they act on tokens. This perspective raises several key questions:

**Q1.** What regularity properties do these trajectories individually exhibit, and how are the blocks related to each other? More concretely, what is the relationship between the block Jacobians across different tokens and transformer layers?

**Q2.** How do the properties of hidden representations and their interactions emerge during training?

**Q3.** Are any of these properties linked to the generalization capabilities of LLMs?

---

[*]Equal contribution. Source code available at https://github.com/sugolov/coupling

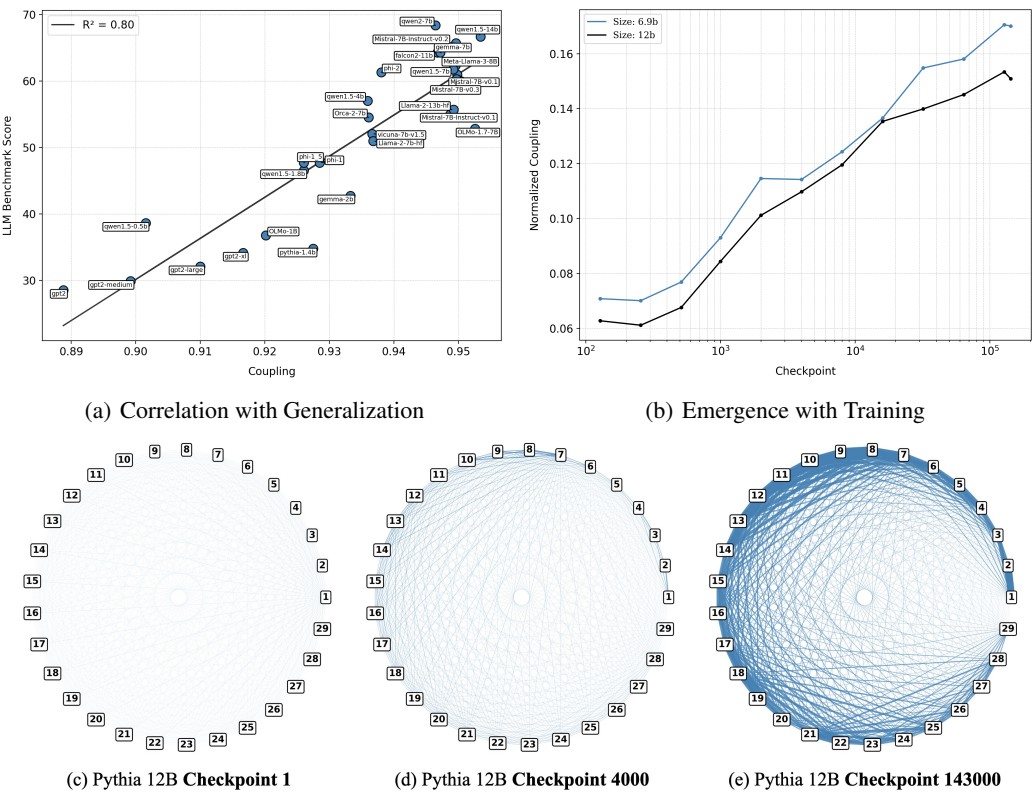

Figure 1: **Transformer Block Coupling Measurements**. **(a)** The plot illustrates the correlation between average coupling (taking $K = \frac{1}{10}d_{\text{model}}$) and benchmark scores across LLMs, showing that higher coupling corresponds to improved performance, with a regression fit yielding an $R^2$ value of 0.8 with a significant p-value of $9.99 \times 10^{-10}$. **(b)** The mean normalized coupling (with $K = 10$) is plotted as a function of training checkpoints for Pythia 12B and 6.9B (Biderman et al., 2023), measured at steps $128, 256, 512, 1k, 2k, \ldots, 128k, 143k$. **(c-e)** Adjacency plots illustrate the mean coupling scores between pairs of layers. Each node represents a layer, and edge weight and opacity indicate the strength of depth-wise normalized coupling. Visualizations are provided for checkpoints 1, 4k, and 143k of Pythia 12B.

## 1.1 CONTRIBUTIONS

We investigate the motivating questions across an extensive collection of openly-available LLMs, most having over 1B parameters, trained by over 8 independent organizations, with varying training methods and data (Appendix A.1). Through our experiments, we identify consistent properties that characterize the **transformer block coupling** phenomenon.

1. **Coupling.** The singular vectors of the Jacobians of the transformer blocks are coupled across depth (Figures 3, 17) and tokens (Figures 4, 15, 16, 18, 19, 20) in several open source LLMs (Table (1)). Furthermore, coupling across Jacobians emerges with training (Figures 1(b), 3, 4, 6(b), 15, 16), and the coupling strength becomes more pronounced between adjacent layers with training, indicating a layer-wise locality in the interactions (Figures 1(c-e)).

2. **Generalization.** The strength of coupling is **correlated with** benchmark performance on the HuggingFace Open LLM Leaderboard (Beeching et al., 2023) (Figures 1(a), 8). Additionally, coupling is more strongly correlated with generalization than parameter budget, model depth, and token embedding dimension (Figure 9). Experiments with ViTs support the relationship observed in LLMs (Figure 6(a)).

3. **Regularity.** Linearity in hidden trajectories emerges with training (Figures 25, 10, 5(a), 12, 23, 24), aligning with behaviour previously observed in ResNets Li & Papyan (2023). Exponential growth occurs in contiguous token representations as a function of depth (Figures 5(b), 22, 23, 26), starkly contrasting linear growth previously observed.

We provide a new perspective on token embedding interactions by examining layers of transformers through their Jacobian matrices. Our results display the effects of training on transformer blocks, and opens potential avenues for promoting generalization in similar architectures.

## 2 BACKGROUND ON LARGE LANGUAGE MODELS

LLMs may be described as a deep composition of functions that iteratively transform token embeddings. In the input layer, $l = 0$, the text prompts are tokenized and combined with the positional encodings to create an initial high-dimensional embedding, denoted by $x_i^0 \in \mathbb{R}^{d_{\text{model}}}$ for the $i^{\text{th}}$ token. When these embeddings are stacked, they form a matrix:

$$X^0 = (x_1^0, x_2^0, \ldots, x_n^0) \in \mathbb{R}^{n \times d_{\text{model}}}. \tag{1}$$

The embeddings then pass through $L$ transformer blocks:

$$X^0 \xrightarrow{F_{\text{block}}^1} X^1 \xrightarrow{F_{\text{block}}^2} \cdots X^{L-1} \xrightarrow{F_{\text{block}}^L} X^L. \tag{2}$$

$X^l = F_{\text{block}}^l(X^{l-1})$ denotes the embeddings after the $l^{\text{th}}$ block, consisting of causal multi-headed attention (MHA), a feed-forward network (FFN), and normalization layers (LN) with residual connections:

$$h^{l+1}(X^l) = \text{MHA}(\text{LN}(X^l)) \tag{3}$$
$$g^{l+1}(X^l) = \text{LN}(X^l + h^{l+1}(X^l)) \tag{4}$$
$$f^{l+1}(X^l) = h^{l+1}(X^l) + \text{FFN}(g^{l+1}(X^l)) \tag{5}$$
$$F_{\text{block}}^{l+1}(X^l) = X^l + f^{l+1}(X^l), \tag{6}$$

where the MHA, LN, FFN are implicitly indexed by layer. Among many models (Appendix 1), an additional rotary positional embedding (RoPE, Su et al. (2023)) is applied in the MHA layer. In the final representation, typically an additional layer normalization is applied:

$$F_{\text{block}}^L(X^{L-1}) = \text{LN}(X^{L-1} + h^L(X^{L-1}) + \text{FFN}(g^L(X^{L-1}))). \tag{7}$$

The output $X^L$ of the final block $F^L$ is passed into a bias-free linear layer $M \in \mathbb{R}^{d_{\text{vocab}} \times d_{\text{model}}}$, with $d_{\text{vocab}}$ denoting the size of the token vocabulary and $d_{\text{model}}$ is the dimension of the token embeddings. This layer $M$ computes final-layer logits for each token embedding, $\ell_i = M x_i^L$. The prediction for the next token is then determined by selecting the maximal logit value: $\arg\max_{v \in \text{tokens}} \ell_{v,n}$.

## 3 METHODS

### 3.1 COUPLING OF SINGULAR VECTORS OF JACOBIANS

**Jacobians.** Coupling is investigated through analyzing the linearizations of transformer blocks which is given by their Jacobian matrices

$$J_{t_1 t_2}^l = \frac{\partial}{\partial x_{t_1}^{l-1}} \left( f^l(X^{l-1}) \right)_{t_2}, \tag{8}$$

defined for each layer $l \in \{1, \ldots, L\}$, and pair of tokens $t_1, t_2 \in \{1, \ldots, n\}$. Note that this is the Jacobian matrix for each transformer block without the contribution of the skip connection from the input of the block, similar to the quantity measured by Li & Papyan (2023) which strictly analyzes the case where $t_1 = t_2$.

Due to the causal structure of the representations, $J_{t_1 t_2}^l = 0$ whenever $t_1 > t_2$. Hence, we restrict our attention to the case where $t_1 \leq t_2$.

**Singular value decomposition (SVD).** We compute the SVD of the Jacobians:

$$J_{t_1 t_2}^l = U_l S_l V_l^\top$$

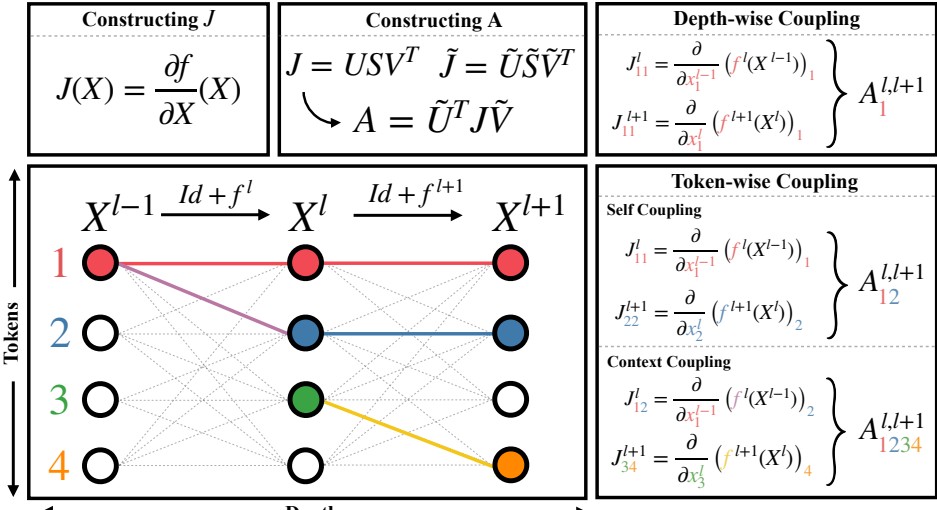

Figure 2: **Transformer Block Coupling.** A visualization of the various types of transformer block coupling with brief instructions on computing both the Jacobians $J$ and coupling matrices $A$ (Section 3.1). The coupling measurement quantifies the alignment and agreement between the interactions of embeddings connections within the network. The colored subscripts in the sample matrices $A$ indicate the specific connections being compared.

where $U_l, V_l \in \mathbb{R}^{d_{\text{model}} \times d_{\text{model}}}$ are the matrices of left and right singular vectors respectively, and $S_l \in \mathbb{R}^{d_{\text{model}} \times d_{\text{model}}}$ contains the singular values.[1]

**Coupling Measurement.** To measure coupling between two Jacobians

$$J_{t_1 t_2}^l = U_l S_l V_l^\top \qquad \text{and} \qquad J_{t_1' t_2'}^{l'} = U_{l'} S_{l'} V_{l'}^\top,$$

we define the *coupling matrix*

$$A_{t_1 t_2 t_1' t_2', K}^{ll'} := U_{l', K}^\top J_{t_1 t_2}^l V_{l', K}$$
$$= U_{l', K}^\top U_l S_l V_l^\top V_{l', K},$$

for $l, l' \in \{1, \dots, L\}$ and $t_1, t_2, t_1', t_2' \in \{1, \dots, n\}$, where $U_{l', K}$ and $V_{l', K}$ are the matrices of top $K$ singular vectors. If the singular vectors of distinct Jacobians are strongly aligned, then the coupling matrix $A$ should be strongly diagonal and approximately equal to the matrix, $S_{l, K}$, of the top $K$ singular values of $J^l$. To quantify the top $K$ mis-coupling, we define

$$m_K(J_{t_1 t_2}^l, J_{t_1' t_2'}^{l'}) = \frac{\|A_{t_1 t_2 t_1' t_2', K}^{ll'} - S_{l, K}\|_F}{\|s_{l, K}\|_p}, \tag{9}$$

where $s_{l, K}$ is the vector of the top $K$ singular values of $J^l$, $\|\cdot\|_F$ denotes the Frobenius norm and $\|\cdot\|_p$ the vector $p$-norm. Typically, we take $p = 1$, which corresponds to normalizing by the sum of the top $K$ singular values of $J^l$. We then define the top $K$ coupling to be $c_K = 1 - m_K$.

We also generalize this analysis to the alignment between left and right singular vectors of different Jacobians by considering the matrix

$$B_{t_1 t_2 t_1' t_2', K}^{ll'} := V_{l', K}^\top J_{t_1 t_2}^l U_{l', K}.$$

**Depth-wise Coupling.** To analyze coupling across transformer blocks, we fix $t$, and measure alignment between $J_{tt}^l$ and $J_{tt}^{l'}$ through the matrix $A_t^{ll'}$ across layers $l, l' \in \{1, \dots, L\}$ [2]

**Token-wise Coupling** We also quantify the coupling across tokens in several ways:

---

[1] Note that the superscripts $t_1, t_2$, indicating the tokens, are omitted for clarity in the expression for the SVD.
[2] In the matrix $A$, we write the single subscript $t$ for clarity.

- **Self-coupling.** By fixing two layers $l, l' \in \{1, \ldots, L\}$, we analyze the case where the input and output tokens are the same. Explicity, we compare $J_{tt}^l$ and $J_{t't'}^{l'}$ across $t, t' \in \{1, \ldots, n\}$, which represents the coupling across tokens for a token's effect on its own trajectory.

- **Context Coupling.** We consider the context tokens' impact on a trajectory by measuring coupling between $J_{t_1 t_2}^l$ and $J_{t_1 t_2'}^{l'}$ across $t_2, t_2' \geq t_1$ (fixing the input token to be the same) and also between $J_{t_1 t_2}^l$ and $J_{t_1' t_2}^{l'}$ across $t_1, t_1' \leq t_2$ (fixing the output token to be the same).

## 3.2 Linearity of Trajectories

Linearity in intermediate embeddings is quantified with the *line-shape score* (LSS), defined by Gai & Zhang (2021) as

$$\text{LSS}_i^{0,\ldots,L} = \frac{L}{\left|\left|\tilde{x}_i^L - \tilde{x}_i^0\right|\right|_2}, \tag{10}$$

where $\tilde{x}_i^0 = x_i^0$, i.e., the input embeddings passed to the LLM, and $\tilde{x}_i^l$ is defined recursively as

$$\tilde{x}_i^l = \tilde{x}_i^{l-1} + \frac{x_i^l - x_i^{l-1}}{\left|\left|x_i^l - x_i^{l-1}\right|\right|_2} \quad \text{for } l = 1, \ldots, L. \tag{11}$$

Note that LSS $\geq 1$, with LSS $= 1$ if and only if the intermediate representations $x_i^0, \ldots, x_i^L$ form a co-linear trajectory.

## 3.3 Layer-Wise Exponential Growth

We measure the presence of exponential spacing (*expodistance*) of the hidden trajectories. Assuming exponential growth of the embedding norms as they flow through the hidden layers, we estimate $\|x_i^l\| \approx e^{\alpha l}\|x_i^0\| = e^\alpha\|x_i^{l-1}\|$ for some fixed $\alpha \in \mathbb{R}$ over all layers $l = 1, \ldots L$. We quantify the validity of this representation by measuring the coefficient of variation of $\alpha_i^l$, given by

$$\alpha_i^l \approx \ln\left(\frac{\|x_i^l\|}{\|x_i^{l-1}\|}\right), \tag{12}$$

for each layer $l$ and token $i$. Under exponential growth, it is expected that $\alpha_i^l$ is independent of depth. We therefore denote the expodistance (ED) of the trajectory of the $i^{th}$ token of a given sequence by

$$\text{ED}_i = \frac{\text{Var}_l \alpha_i^l}{(\text{Avg}_l \alpha_i^l)^2}. \tag{13}$$

This measurement is motivated by the discussion in Section 6.1, the parametrization in Appendix A.5, and empirical evidence from Figure 27a. It also serves as a method to assess the validity of the linearization presented in Equation 14.

## 3.4 Vision Transformer Training

For further investigation of coupling in transformers, we consider Vision Transformers (ViTs) following DEiT (Touvron et al., 2021). We train 3 ViTs for classification on CIFAR10 (Krizhevsky, 2009) with varied stochastic depth rate for embedding size 192, depth 24, and 3 attention heads. Please see Appendix A.7 for further details.

## 4 Evaluation

### 4.1 Suite of Large Language Models

Our study evaluates a comprehensive collection of LLMs (see Appendix A.1) that were independently trained by various individuals and organizations. These models, provided through Hugging-Face (Wolf et al., 2020), vary in terms of parameter budgets, number of layers, hidden dimensions,

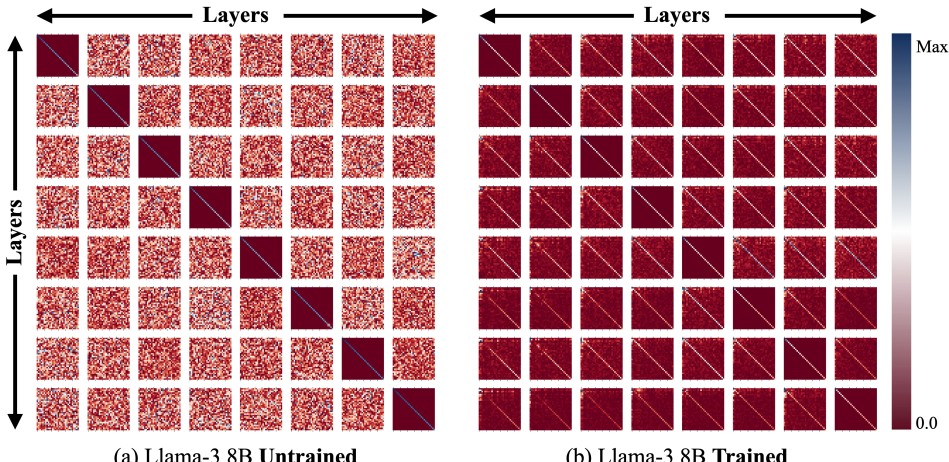

(a) Llama-3 8B **Untrained**  (b) Llama-3 8B **Trained**

Figure 3: **Transformer Block Coupling across Depth**. The figure shows Jacobian coupling across transformer blocks 9 to 16, using the prompt "What is the capital of France? The capital is" to trace the final token's trajectory. In trained models (bottom row), the diagonal pattern with minimal off-diagonal values indicates alignment of Jacobians, where top singular vectors of $J^{l'}$ diagonalize $J^l$. Untrained models do not exhibit such coupling. Further details are in the Appendix A.8 (Figure 17). Best viewed in color.

and training tokens. Moreover, we analyze the dynamics of each measurement throughout training by deploying the Pythia Scaling Suite (Biderman et al., 2023). A summary of the models under consideration is presented in Table 1 of Appendix A.1 and further details in Appendix A.6.

## 4.2 PROMPT DATA

We evaluate these LLMs using prompts of varying length, ambiguity, and context, sourced from the test set of ARC (Clark et al., 2018), GSM8K (Cobbe et al., 2021), HellaSwag (Zellers et al., 2019), MMLU (Hendrycks et al., 2021), Truthful QA (Lin et al., 2022), and Winogrande (Sakaguchi et al., 2019). These set the performance benchmarks on the HuggingFace Open LLM Leaderboard (Beeching et al., 2023) and provide a representative evaluation of performance on many language tasks.

## 5 RESULTS

### 5.1 COUPLING OF JACOBIANS ACROSS DEPTH

In trained LLMs, we observe coupling of the top singular vectors of the Jacobians across depth (Figure 3 bottom row), evident in the low non-diagonal values with a visible diagonal present in the matrix subplots. This is consistently observed across various LLMs considered. On the other hand, in untrained models (Figure 3 top row), there is no coupling of Jacobians across different depths. There is coupling along the diagonal, however, because each Jacobian is trivially diagonalized by its own singular vectors. This, in addition to Figure 1(b), suggests that coupling across depth emerges through training.

### 5.2 COUPLING OF JACOBIANS ACROSS TOKENS

We analyze the coupling of singular vectors of Jacobians across tokens. For input and output tokens that are the same ($J_l^{tt}$ and $J_{l'}^{t't'}$, Figure 4), we observe strong coupling, indicating that a token's interactions along its trajectory are coupled with others. For context tokens, coupling is examined by fixing the input token ($J_l^{t_1t_2}$ and $J_{l'}^{t_1t_2'}$, Figure 15) or the output token ($J_l^{t_1t_2}$ and $J_{l'}^{t_1't_2}$, Figure

16). While context coupling exists, its strength varies across token pairs. Untrained models show no such coupling with randomly fixed layers $l, l'$ in general.

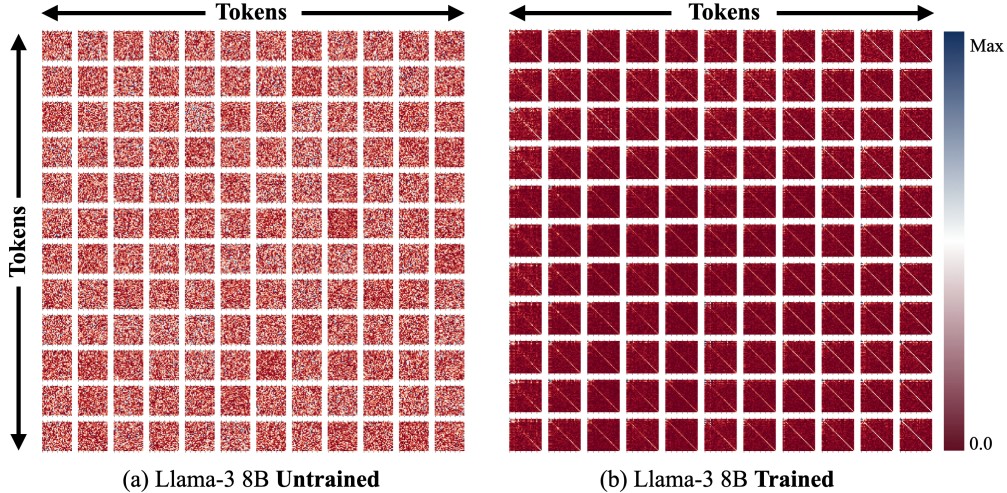

(a) Llama-3 8B **Untrained**   (b) Llama-3 8B **Trained**

Figure 4: **Transformer Block Coupling across Tokens (Self Coupling)**. The figure shows Jacobian coupling for the same input and output token across tokens, visualized using the absolute values of $A_{ll'}^{ttt't'}$ (with fixed layers $l, l'$). In trained models (bottom row), the strong diagonal and small off-diagonal values indicate coupling, while no such coupling is present at initialization (top row). Additional plots are in Appendix A.8 (Figure 18).

### 5.3 EMERGENCE OF COUPLING WITH TRAINING

Coupling emerges through training for the evaluated LLMs, including coupling across depth (Figure 3) and across tokens (Figures 4, 15, 16). Further, we evaluate layer-wise coupling at intermediate training checkpoints of Pythia 6.9B and 12B (Biderman et al., 2023) Figure 1(b), and observe that coupling is generally low at initialization and increases persistently throughout training. Moreover, there is a clear sense of locality in the strength of coupling which is visually displayed in Figure 1c-e)

The gradual growth of coupling observed in Figure 1(b) parallels the logarithmic increase in accuracy during training for Pythia 12B and Pythia 6.9B (Biderman et al., 2023). This highlights the relationship between coupling and performance, since both properties emerge at similar training iterations and rates.

### 5.4 COUPLING IN VISION TRANSFORMERS

To further investigate coupling, we train several ViTs on CIFAR-10. Our experiments confirm that coupling emerges with training (Figure 6(b)) and exhibits a strong positive correlation with test accuracy (Figure 6(a)). These findings suggest that coupling is a general phenomenon in transformers and reinforces our observations in LLMs. Notably, we measure coupling in ViTs on train data while its correlation with accuracy persists on test data, highlighting coupling as a structural property of the model related to generalization.

### 5.5 CORRELATION WITH GENERALIZATION

For each LLM, we measure the average coupling across depth for prompts in the 6 evaluation datasets (Section 4.2), where for each prompt, $c_K(J_n^l, J_n^{l'})$ is averaged over layers $l, l' \in \{1, \ldots, L\}$. We then plot the coupling values against the benchmark scores across several LLMs (Figure 1(a)). Our results reveal a positive correlation between coupling and performance benchmark scores, exceeding the significance of correlations with other model hyperparameters (Figure 9). This observa-

tion suggests a compelling relationship between stronger coupling of singular vectors of Jacobians $J_l^{t_1 t_2}$ and improved generalization.

We hypothesize that simple trajectories may lead to better generalization. This idea aligns with with several generalization bounds in machine learning (Arora et al., 2018), which suggest that models with lower complexity tend towards better generalization. Furthermore, previous work (Novak et al., 2018) demonstrated that the Frobenius norm of input-output Jacobians is related to generalization, suggesting that coupling, a structural property derived from block Jacobians, may also be related to generalization.

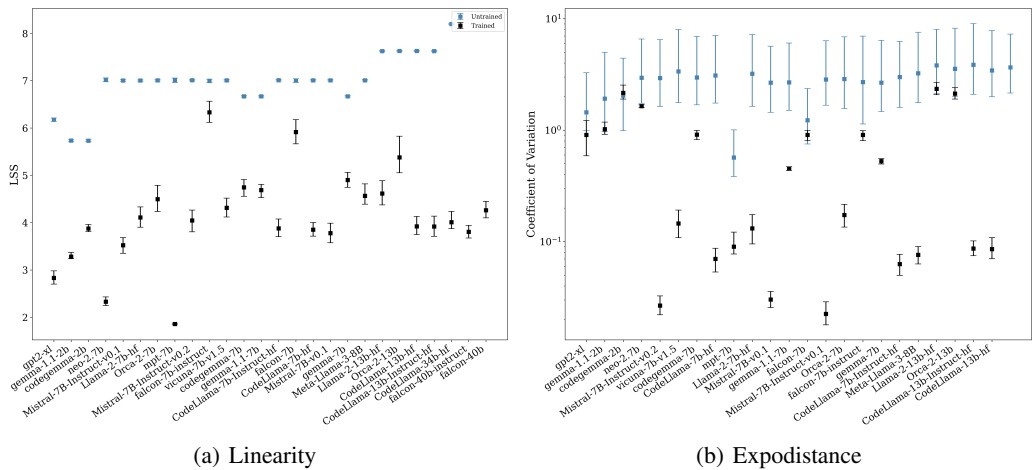

(a) Linearity  (b) Expodistance

Figure 5: **Regularity of Trajectories.** The line-shape score (LSS) of embedding trajectories, as discussed in Section 5.6, computed on 1,200 prompts of the HuggingFace Open LLM Leaderboard (Section 4.2) for a variety of trained (black) and randomly initialized (blue) LLMs (Appendix A.1). Median values over all prompts are plotted and are accompanied with uncertainty intervals depicting the inter-quartile range of the results for each model. Models are sorted by number of parameters.

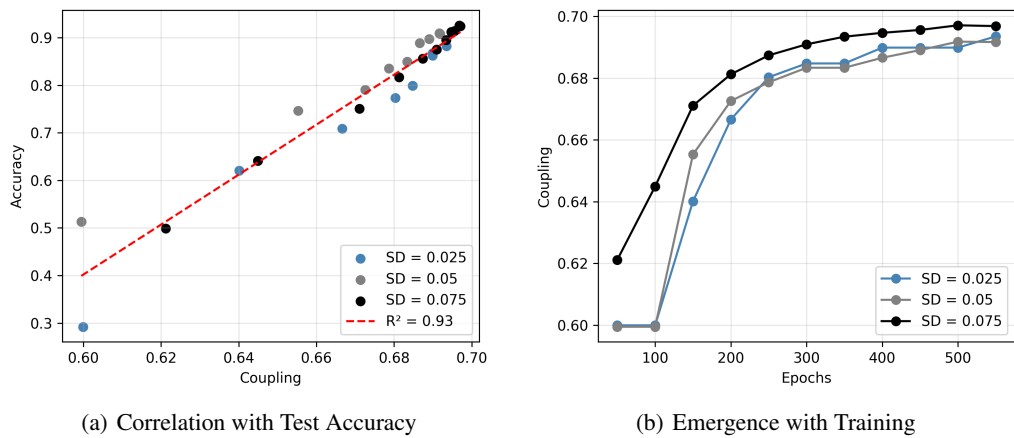

(a) Correlation with Test Accuracy  (b) Emergence with Training

Figure 6: **Transformer Block Coupling in ViTs.** (a) CIFAR10 test accuracy against coupling for stochastic depths $(0.025, 0.05, 0.075)$ (Section 3.4). Coupling and accuracy are evaluated at training checkpoints (see (b)), showing a positive correlation ($R^2 = 0.93$). (b) Coupling at training checkpoints $\{50, 100, \ldots, 550\}$ for transformers trained with stochastic depths $(0.025, 0.05, 0.075)$.

### 5.6 REGULARITY IN EMBEDDING TRAJECTORIES

By analyzing token trajectories as representations propagate through the layers, we identify several structural properties. Notably, we observe a considerable degree of linearity in the hidden trajectories across all examined LLMs. The LSS of the trajectories has an average value $4.25$ in trained models, compared to $6.54$ at initialization, and is supported by the low variation across benchmark prompts (Figure 5). Linearity increases with training at varying depths of Pythia 12B (Figure 25). This linear and expansive behavior of the representations is further illustrated in Llama-3 70B, MPT 30B, and NeoX 20B through low dimensional projections of embedding trajectories (Figure 10).

Among the LLMs studied, most hidden trajectories exhibit exponential growth that emerges with training (Figure 5(b)). The low coefficient of variation across prompts indicates the robustness of this property across diverse tasks. In contrast, measurements at initialization show equally (rather than exponentially) distanced trajectories, as reflected by the low coefficients of variation in the norms of their layer-wise differences (Figure 22). Under certain assumptions, exponential spacing is motivated through coupling, as discussed in in Section 6.1 and Appendix A.5.

## 6 DISCUSSION

### 6.1 EMERGENCE OF REGULARITY WITH COUPLING

Under certain assumptions, the increased linearity and exponential spacing observed in many models can be analyzed as a consequence of coupling. Considering input embeddings $x_1^0, \ldots, x_n^0$ and the linearization of the last token embedding $x_n^l$ given by $J_{n,n}^l(x_1^0, \ldots, x_n^0)$:

$$x_n^l = (I + J_{n,n}^l(x_1^0, \ldots, x_n^0))x_n^{l-1}.$$

To simplify notation, we write $x_n^l = x^l$, $J_{n,n}^l(x_1^0, \ldots, x_n^0) = J^l$. The representations follow the linearized equation:

$$x^{l+1} = x^l + J_l x^l = (I + J_l)x^l.$$

Expanding across layers, the entire system can be approximated by the product

$$x^L = (I + J_L)(I + J_{L-1})\cdots(I + J_1)x^0.$$

Assuming perfect coupling of singular vectors across depth (and between left and right singular vectors), we have $J_l \approx U S_l U^T$. As discussed in Appendix A.5, assuming the singular values across depth are approximately constant ($S_l = S$), this equation predicts that the norm of $x_l$ would exhibit exponential growth if $x_0$ is some eigenvector of $J$. Expanding $U$ and $S$,

$$x^l = \sum_{j=1}^{d_{\text{model}}} u_j (1 + s_j)^l u_j^\top x^0.$$

where $u_j$ and $s_j$ represent the singular vectors and singular values of the Jacobian, respectively. In general, trajectories are not expected to be perfectly linear unless $x^0$ aligns with an eigenvector of $J$. However, in our experiments, we observe a notable tendency towards linearity, suggesting that the representations align progressively during training with the eigenvectors of the coupled Jacobians.

## 7 RELATED WORK

**Residual Networks.** ResNets (He et al., 2016) have been viewed as an ensemble of shallow networks (Veit et al., 2016), with studies delving into the scaling behaviour of their trained weights (Cohen et al., 2021). The linearization of residual blocks by their Residual Jacobians was first explored by Rothauge et al. (2019), who examined Residual Jacobians and their spectra in the context of stability analysis, and later by Li & Papyan (2023) who discovered Residual Alignment. Coupling of Jacobian singular vectors in LLM transformers extends previous results for Resnets (Li & Papyan, 2023). We show coupling in $J_l^{t_1 t_2}$ across various tokens, which is specific to tokenization in transformers, in addition to demonstrating coupling of $J_l^{t_1 t_2}$ across $l$, which was also identified analogously in ResNets (Li & Papyan, 2023). Further comparison is included in Appendix A.3.

**Neural Ordinary Differential Equations.** Neural ODEs (Chen et al., 2018) view ResNets as a discretized dynamical process, with past work (Sander et al., 2022) showing the convergence of Residual Networks to a system of linear ODE, with some extensions to transformers (Zhong et al., 2022; Li et al., 2021) The emergence of coupling in transformers suggests that a discretization of a simple iterative process emerges in LLMs.

**In-context learning.** LLMs can perform tasks through examples provided in a single prompt, demonstrating in-context learning (von Oswald et al., 2023; Bai et al., 2024; Ahn et al., 2023; Akyürek et al., 2023; Xie et al., 2021; Hahn & Goyal, 2023; Xing et al., 2024). Studies suggest trained self-attention layers implement gradient-descent-like updates across depth to minimize the MSE of a linear model:

$$x_{\text{final}} = \min_x ||Ax - b||^2.$$

These updates take the form:

$$x_{t+1} = (I + \epsilon A^T A)x_t - \epsilon A^T b.$$

Coupling across depth suggests similarity in the matrices $I + \epsilon A^T A$.

**Hidden Representation Dynamics.** Prior research interprets deep neural networks through dynamical systems, revealing that training trajectories align with geodesic curves (Gai & Zhang, 2021) and partition activation space into basins of attraction (Nam et al., 2023). For further related works, see (Geshkovski et al., 2023b;a; Tarzanagh et al., 2023; Valeriani et al., 2023; Hosseini & Fedorenko, 2023).

**Structure in Hidden Representations.** Neural Collapse (Papyan et al., 2020) highlights emergent regularity in last-layer representations, with subsequent studies exploring hidden-layer structures and their theoretical underpinnings (Wang et al., 2024a; Parker et al., 2023; Zangrando et al., 2024; Garrod & Keating, 2024; Wang et al., 2024b; Hoyt & Owen, 2021; Arous et al., 2023; Zarka et al., 2021; Ben-Shaul & Dekel, 2022; Papyan, 2020; Súkeník et al., 2023). In LLMs, recent works identify uniform token structures (Wu & Papyan, 2024) and low-dimensional hidden trajectories (Sarfati et al., 2024). Our work examines local token interactions through Jacobian dynamics across all layers.

## 8 CONCLUSION

Our primary goal was to contribute to the understanding of the mechanics underlying transformer architectures through an analysis of the trajectories of token embeddings and their interactions. Our research builds on the understanding of transformer architectures by revealing the coupling, across depth and token, of singular vectors in the Jacobians of transformer blocks for multiple LLMs trained by various organizations, as well as ViTs. We establish a correlation between the strength of this coupling and benchmark performance on the HuggingFace Open LLM Leaderboard, highlighting the significance of transformer block coupling for generalization. These findings open avenues for future research, encouraging deeper exploration into the connections between regularity of hidden representations, training methods, and generalization.

ACKNOWLEDGMENTS

We acknowledge the support of the Natural Sciences and Engineering Research Council of Canada (NSERC). This research was supported in part by the Province of Ontario, the Government of Canada through CIFAR, and industry sponsors of the Vector Institute (`www.vectorinstitute.ai/partnerships/current-partners/`). This research was also supported in part by Compute Ontario (`https://www.computeontario.ca/`) and the Digital Research Alliance of Canada (`https://alliancecan.ca/`).

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

## A   APPENDIX AND SUPPLEMENTARY MATERIAL

### A.1   SUITE OF LARGE LANGUAGE MODELS AND PROMPT DATA

We evaluate a comprehensive collection of LLMs on 6 datasets from the HuggingFace Open LLM leaderboard (Beeching et al., 2023).

**Base models.** Falcon (40B, 7B) (Almazrouei et al., 2023), Falcon2 (11B) (Malartic et al., 2024), Llama-3 (70B, 8B) (AI@Meta, 2024), Llama-2 (70B, 13B, 7B) (Touvron et al., 2023), MPT (30B, 7B) (Team, 2023a;b), Mistral v0.1 (7B) (Jiang et al., 2023), Gemma (7B, 2B) (Team et al., 2024), Gemma 1.1 (7B, 2B), Phi (1B, 1.5B, 2B) (Gunasekar et al., 2023; Li et al., 2023), OLMo (1B, 7B) (Groeneveld et al., 2024), Qwen (Bai et al., 2023; Yang et al., 2024), NeoX (20B) (Black et al., 2022), Neo (2.7B) (Black et al., 2021; Gao et al., 2020), Pythia (1.4B, 2.8B, 6.9B, 12B) (Biderman et al., 2023), and GPT-2 (1.5B, 774M, 355M, 117M) (Radford et al., 2019).

**Fine-tuned models.** CodeLlama (34B, 13B, 7B) (Rozière et al., 2024), CodeLlama Instruct (34B, 13B, 7B) (Rozière et al., 2024), Mistral-v0.1 Instruct (7B) (Jiang et al., 2023), Mistral-v0.2 Instruct (Jiang et al., 2023), CodeGemma (Team et al., 2024).

**Datasets.** ARC (Clark et al., 2018), HellaSwag (Zellers et al., 2019), MMLU (Hendrycks et al., 2021), Truthful QA (Lin et al., 2022), WinoGrande (Sakaguchi et al., 2019).

Table 1: **LLMs featured in the experiments throughout paper.** Included in the table is the parameter budget of each model, the embedding dimension, the number of training tokens, and the Open LLM leaderboard (Beeching et al., 2023) benchmark score.

| MODEL | PARAM. | LAYERS ($L$) | DIM. ($d_{\mathrm{MODEL}}$) | TOKENS | SCORE | POS.ENCODING |
|---|---|---|---|---|---|---|
| LLAMA-3 | 70 B | 80 | 4096 | 15 T | | RoPE |
| | 8 B | 32 | 4096 | 15 T | 62.35 | RoPE |
| LLAMA-2 | 70 B | 80 | 8192 | 2 T | 66.05 | RoPE |
| | 13 B | 40 | 5120 | 2 T | 55.69 | RoPE |
| | 7 B | 32 | 4096 | 2 T | 50.97 | RoPE |
| CODELLAMA | 34 B | 48 | 8192 | 2 T | 55.33 | RoPE |
| | 13 B | 40 | 5120 | 2 T | 45.82 | RoPE |
| | 7 B | 32 | 4096 | 2 T | 39.81 | RoPE |
| CODELLAMA (IT) | 34 B | 48 | 8192 | 2 T | 43.0 | RoPE |
| | 13 B | 40 | 5120 | 2 T | 37.52 | RoPE |
| | 7 B | 32 | 4096 | 2 T | 40.05 | RoPE |
| ORCA 2 | 13 B | 40 | 5120 | 2 T | 58.64 | RoPE |
| | 7 B | 32 | 4096 | 2 T | 54.55 | RoPE |
| FALCON | 40 B | 60 | 8192 | 1 T | 58.07 | RoPE |
| | 7 B | 32 | 4544 | 1.5 T | 44.17 | RoPE |
| FALCON (IT) | 40 B | 60 | 8192 | 1 T | 43.26 | RoPE |
| | 7 B | 32 | 4544 | 1.5 T | | RoPE |
| FALCON2 | 11 B | 60 | 4096 | | 64.28 | RoPE |
| MISTRAL-V0.1 | 7 B | 32 | 4096 | | 60.97 | ALiBi |
| MISTRAL-V0.1 (IT) | 7 B | 32 | 4096 | | 54.96 | ALiBi |
| MISTRAL-V0.2 (IT) | 7 B | 32 | 4096 | | 65.71 | ALiBi |
| MISTRAL-V0.3 | 7 B | 32 | 4096 | | 60.28 | ALiBi |
| VICUNA | 7 B | 32 | 4096 | | 52.06 | RoPE |
| MPT | 30 B | 48 | 7168 | 1 T | 66.98 | ALiBi |
| | 7 B | 32 | 4096 | 1 T | 56.84 | ALiBi |
| OLMO | 7 B | 32 | 4096 | 2.5 T | 43.36 | RoPE |
| | 1 B | 16 | 2048 | 3 T | 36.78 | RoPE |
| OLMO-1.7 | 7 B | 32 | 4096 | 2.5 T | 52.82 | RoPE |
| PHI-2 | 2 B | 32 | 2560 | 1.4 T | 61.33 | RoPE |
| | 1.5 B | 24 | 2048 | 150 B | 47.69 | RoPE |
| | 1 B | 24 | 2048 | 54 B | 47.69 | RoPE |
| GEMMA | 7 B | 28 | 3072 | 6 T | 64.29 | RoPE |
| | 2 B | 18 | 2048 | 6 T | 42.75 | RoPE |
| GEMMA-1.1 | 7 B | 28 | 3072 | 6 T | 60.09 | RoPE |
| | 2 B | 18 | 2048 | 6 T | 30.0 | RoPE |
| CODEGEMMA | 7 B | 28 | 3072 | 6 T | 56.73 | RoPE |
| | 2 B | 18 | 2048 | 6 T | 32.19 | RoPE |
| QWEN1.5 | 14 B | 40 | 5120 | | 66.7 | RoPE |
| | 7 B | 32 | 4096 | | 61.76 | RoPE |
| | 4 B | 40 | 2560 | | 57.05 | RoPE |
| | 1.8 B | 24 | 2048 | | 46.55 | RoPE |
| | 0.5 B | 24 | 1024 | | 38.62 | RoPE |
| QWEN2 | 7 B | 28 | 3584 | | 68.4 | RoPE |
| NEO | 20 B | 44 | 6144 | | 41.69 | RoPE |
| | 2.7 B | 32 | 2560 | 0.42 T | 36.20 | SINE |
| PYTHIA | 12 B | 36 | 5120 | 0.3 T | 38.82 | RoPE |
| | 6.9 B | 32 | 4096 | 0.3 T | 39.30 | RoPE |
| | 2.8 B | 32 | 2560 | 0.3 T | 37.09 | RoPE |
| | 1.4 B | 24 | 2048 | 0.3 T | 34.75 | RoPE |
| | 1 B | 16 | 2048 | 0.3 T | 32.78 | RoPE |
| GPT-2 | 1.5 B | 48 | 1600 | | 34.12 | SINE |
| | 774 M | 36 | 1280 | | 32.07 | SINE |
| | 355 M | 24 | 1024 | | 29.87 | SINE |
| | 117 M | 12 | 768 | | 28.53 | SINE |

## A.2 ADDITIONAL METRICS

### A.2.1 VISUALIZATION OF TRAJECTORIES WITH PCA

Each token, with initial embedding $x_i^0$, forms a trajectory $x_i^0, x_i^1, \ldots, x_i^L$ as it passes through the $L$ transformer blocks. The dynamics in high-dimensional space are visualized through a 2-dimensional principal component (PC) projection, $\mathrm{PC}_L$, fitted to the last layer embeddings $X^L = (x_1^L, x_2^L, \ldots x_n^L)$. The projected embeddings, $\mathrm{PC}_L(x_i^0)$, $\mathrm{PC}_L(x_i^1)$, $\ldots$, $\mathrm{PC}_L(x_i^L)$, are plotted for each of the $i = 1, \ldots, n$ trajectories.

## A.3 COMPARISON TO RESNETS

**Coupling.** Our results complement and build upon those of (Li & Papyan, 2023), who have observed the coupling of singular vectors of Residual Jacobians in classification ResNets. We observe coupling across depth in a wide range of LLMs, where Jacobians are evaluated at the sequence of embeddings, with respect to the current token, whereas in RA Jacobians are evaluated at a single representation. Additionally, with transformers we may analyze coupling not only across depth but also across tokens. We observe coupling in LLMs across tokens in a variety of ways. Further, we consider and analyze the relationship between coupling and generalization.

**Linearity and Equidistance.** Linearity in hidden trajectories, as observed in ResNets Li & Papyan (2023); Gai & Zhang (2021), also emerges with training in LLMs. The mean LSS value among the evaluated LLMs is $4.24$ (Figure 5(a)), greater than LSS measurements observed for ResNets (Gai & Zhang (2021), page 18) which range between 2.0-3.0 (due to varying trajectory length and hidden dimension). In both architectures, training induces improved linearity and regularity (Figure 10) in trajectories. In contrast to ResNets, trajectories are not equidistant, instead showing exponential growth between layers (Figure 5(b)). We quantify this spacing through a low coefficient of variation, displaying the presence of exponential growth in token trajectories. In both classifier ResNets and LLM transformers, there is an evident level of regularity in hidden representations (Figure 10).

**Rank of Jacobians.** Li & Papyan (2023) show that Residual Jacobians have rank at most $C$, the number of classes. This analogous result automatically holds for LLMs since the vocabulary size is significantly greater than the embedding dimension of the transformer blocks.

**Singular Value Scaling.** Li & Papyan (2023) observe that top singular values of Residual Jacobians scale inversely with depth. In trained LLMs, however, top singular values do not show a consistent depth scaling across models (Figure 30), notably differing from classification ResNets. In addition, the distribution of singular values at each layer varies significantly between models. Singular value scaling is more present in untrained transformers (Figure29), likely caused by additional layer normalizations in residual blocks (Section 4.1).

## A.4 SIGNIFICANCE OF COUPLING

The transformer block coupling phenomenon offers insight into several prominent practices in LLM research, as summarized in Table 2.

The coupling phenomenon provides insight into the internal operations of transformer. We hypothesize that during training, the LLM learns to represent embeddings in specific low-dimensional subspaces (Eldar & Mishali, 2009). Given an input, the first layer converts the input into embeddings within one of these learned subspaces. Each subsequent transformer layer modifies these embeddings, potentially moving them to different subspaces. Strong coupling between consecutive layers suggests that the LLM tends towards representations in the same or similar subspaces across many layers. Weak coupling suggests that the subspaces may change between layers, though usually gradually, and that adjacent layers still operate in relatively similar subspaces. Previous works have shown (Lad et al., 2024; Gromov et al., 2024) that the early and late layers of language models behave differently, which may be understood through coupling; tokens remain in similar spans, then transition to a different subspace, continuing within a new span that is consistent in the remainder of the transformer.

The emergence of coupling with training steps (Figure 1) may provide insight into the dynamics. Under full coupling and a difference equation approximation, the representations evolve as

$$x^l = \sum_{j=1}^{d_{\text{model}}} u_j (1 + s_j)^l u_j^\top x^0$$

where $u_i$ and $\lambda_i$ denote the eigenvectors and eigenvalues of the Jacobian, respectively. In this case, the gradients of the loss $L$ with respect to prediction $y$, $x^L$ are represented by

$$\frac{\partial L}{\partial x^0}(x^L, y) \approx \sum_{j=1}^{d_{\text{model}}} u_j (1 + s_j)^L u_j^\top (y - x^L),$$

Due to the coupling, the dynamics exhibit either exponential growth or decay in different subspaces, depending on whether $s_j$ is greater or less than 1, which is known to cause challenges for optimization as in past work on dynamical isometry (Pennington et al., 2017). We infer that increasing coupling during training makes optimization progressively more difficult. Conversely, as training progresses, it becomes harder to achieve stronger coupling, and is consistent with the logarithmic trend in Figure 1.

Table 2: **Significance of the Coupling Phenomenon.** A table which highlights the implications of transformer block coupling to a variety of effors in machine learning research.

| Current Research Practice | Key Idea | Our Contribution |
|---|---|---|
| Compressing models by merging blocks (Fu et al., 2022; Kim et al., 2024) | Combine adjacent transformer blocks to reduce model size without significant performance loss. | Demonstrates that merging is effective because blocks become strongly coupled during training. |
| Compressing models by pruning blocks (Elkerdawy et al., 2020; Kim et al., 2024; Dror et al., 2021; Fang et al., 2023) | Remove certain transformer blocks while preserving functionality. | Explains that pruning works because the coupling ensures redundancy across blocks. |
| Compressing models by projecting weight matrices (Ashkboos et al., 2024) | Reduce dimensionality by projecting weights into smaller subspaces. | Shows that coupling induces a low-dimensional subspace in which blocks' weights are aligned. |
| Studying the effect of transformer block permutations (Hu et al., 2021; Mahabadi et al., 2021; van der Ouderaa et al., 2024; Li et al., 2016) | Investigate whether permuting the order of blocks affects model performance. | Explains why permutations have minimal impact: strong coupling creates structural robustness. |
| Early exiting in LLMs (Scardapane et al., 2020; Jazbec et al., 2024) | Allow models to exit computation early based on task confidence. | Reveals that early exiting works because representations progress linearly along a shared trajectory due to coupling. |

### A.5 DYNAMICAL MOTIVATION

The equality of top left and right singular vectors suggests that the linearizations form a simple linearized system that acts on representations. Consider a difference equation

$$x^{l+1} - x^l = A_l x^l \tag{14}$$

Its solution at the final $L$ is given by

$$x^L = \prod_{l=1}^{L} (I + A_l) x^0 \tag{15}$$

Expanding the brackets shows that $x^l$ can be thought of as a collection of many paths of various lengths, due to the binomial identity. This agrees with Veit et al. (2016) which views ResNets as

$$x^l = (I + A_{l-1})(I + A_{l-2})\dots(I + A_1)x^0 \tag{16}$$

However, Veit et al. (2016) do not make any assumptions about the alignment of the various $A_l$ matrices. The coupling phenomenon suggests the model as implementing the simpler system

$$x^l = (I + A)^l x^0 \tag{17}$$

where all the $A$ matrices are aligned. One benefit of this interpretation is that, we can write $x^l$ in a simple closed form, as above. We quantify the similarity of hidden trajectories to the evolution of the above difference equation, in order to detect the emergence of a simple linearization to representations. The emergence of increased linearity and exponential spacing in many LLMs can be analyzed as a result of coupling under some conditions on the spectral decomposition of the Jacobians. Consider input embeddings $x_1^0, \dots, x_n^0$ and the linearization of the last token embedding $x_n^l$ given by $J_{n,n}^l(x_1^0, \dots, x_n^0)$:

$$x_n^l = (I + J_{n,n}^l(x_1^0, \dots, x_n^0))x_n^{l-1}. \tag{18}$$

We simplify notation and write $x_n^l = x^l$, $J_{n,n}^l(x_1^0, \dots, x_n^0) = J^l$. Under the assumption of spectral coupling, $J^l = U_l S_l V_l^T \approx U S_l U^T$, and the linearized effect of the last token is

$$x^L = \prod_{l=1}^L (I + U_l S_l V_l^T)x^0 \approx U\left(\prod_{l=1}^L (I + S_l)\right)U^T x^0 \tag{19}$$

Suppose that $x^0 = u_k$ is the $k$-th left singular vector of $J^l$. It follows that $x^L = \prod_{l=1}^L (1 + s_k^l)x^0$, where $s_k^l$ denotes the $k$-th singular value at layer $l$. The exponential spacing measurement is motivated by the consistent choice $s_k = s_k^1 = s_k^2 = \dots = s_k^L$. Explicitly, under such an assumption, $x^L = (1 + s_k)^L x^0$, and by Equation 12, for each $l$

$$\alpha_k^l = \ln\left(\frac{(1 + s_k)^l ||u_0||}{(1 + s_k)^{l-1}||u_0||}\right) = \ln(1 + s_k) \implies \text{ED} = 0,$$

that is, the coefficient of variation 0 across $l$. In addition, if $x^l = (1 + s_k)x^{l-1}$, it is clear that the trajectory would form a perfect line, yielding $LSS = 1$ by the discussion in Section 3.2. In general, trajectories are not expected to be perfectly linear unless $x^0$ aligns with an eigenvector of $J^l$.

## A.6 LLM Evaluation and Implementation Details

The source code used to produce the results reported in this experiment has been included as supplemental material. Models with varying parameter sizes are loaded on GPUs with appropriate memory requirements: NVIDIA A40 ($n_{\text{param}} \geq 40B$), NVIDIA Quadro RTX 6000 for Gemma variants and when ($40B > n_{\text{param}} > 13B$), and NVIDIA Tesla T4 when ($13B \geq n_{\text{param}}$) except Gemma variants. 1,200 prompts from the OpenLLM leaderboard were evaluated in variable batch sizes were queued on a SLURM cluster, with appropriate adjustments depending on the memory required to load the LLM.

- $13B \geq n_{\text{param}}$: 100 prompts per batch, except Gemma variants, which used 25 prompts per batch. The larger memory requirement for Gemma variants is likely due to the much larger vocabulary size in the model.
- $40B > n_{\text{param}} > 13B$: 10 prompts per batch, except NeoX 20B which used 100 prompts per batch.
- $n_{\text{param}} \geq 40B$: 50 prompts per batch.

Due to the high memory requirement for computing block Jacobians, for experiments involving the Jacobians, NVIDIA Quadro RTX 6000 was used additionally for $13B > n_{\text{param}} \geq 7B$ and corresponding models were quantized. Additionally, coupling of singular vectors of Jacobians was computed on a smaller subset of the dataset prompts.

The computational complexity for the metrics utilized in this paper are as follows:

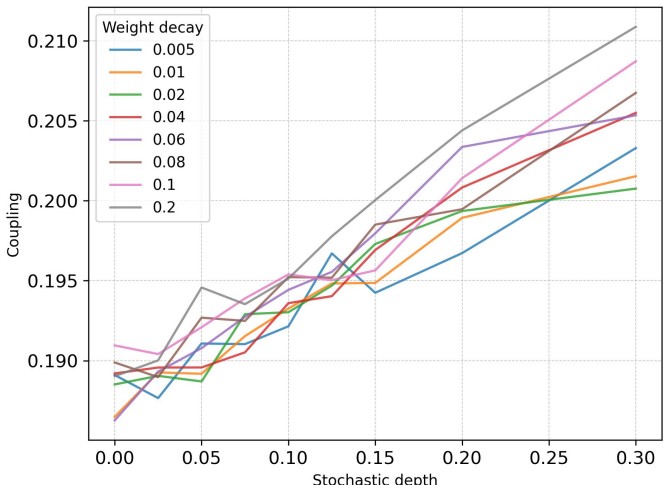

Figure 7: **Coupling against Stochastic Depth Rate.** Plots are generated for each weight decay in $\{0.005, 0.01, 0.02, 0.04, 0.06, 0.08, 0.1, 0.2\}$.

**Coupling.** Coupling requires computing the Jacobians for each transformer block, and so a forward pass and backward pass required (note that we compute the Jacobians on a block level). Once the Jacobians are obtained, it requires computing a truncated singular value decomposition of each Jacobian. The time complexity of computing the truncated SVD of rank $k$ for a $d \times d$ matrix is $\mathcal{O}(d^2k)$, where $k << d$. Computing $A$ from the SVDs then has time complexity $\mathcal{O}(k^3)$, so the asymptotic time complexity of computing the coupling score between two connections is $\mathcal{O}(d^2k)$.

**LSS.** For each trajectory, the time complexity of computing the LSS is $\mathcal{O}(Ld)$ where $L$ is the number of layers and $d$ is the hidden dimension. Therefore, for a prompt containing $T$ tokens, the total time complexity for each prompt is $\mathcal{O}(TLd)$ (in addition to a single forward pass of the model).

**Expodistance.** Similarly, computing the expodistance of a single trajectory has time complexity $\mathcal{O}(Ld)$. Therefore, for a prompt containing $T$ tokens, the total time complexity for each prompt is $\mathcal{O}(TLd)$ (in addition to a single forward pass of the model).

## A.7    ViT TRAINING DETAILS

For further investigation of coupling in transformers, we train 3 Vision Transformers (ViTs) following the default configurations of DEiT training (Touvron et al., 2021) on CIFAR10 (Krizhevsky, 2009). For a fixed ViT architecture with embedding dimension 192, depth of 24 layers, and 3 attention heads, we vary the weight decay 0.05 and stochastic depth rate $\{0.025, 0.05, 0.075\}$. Optimization uses ADAM optimizer with $5e - 3$ learning rate and cosine scheduler for 600 epochs with 10 epochs of linear warmup. Training proceeds with data mixup using $\alpha = 0.8$ Optimization proceeds on 4 NVIDIA Tesla T4 GPUs with 128 batch size and data parallelization, with total training time being approximately 6 hours.

## A.8    ADDITIONAL EXPERIMENTAL RESULTS

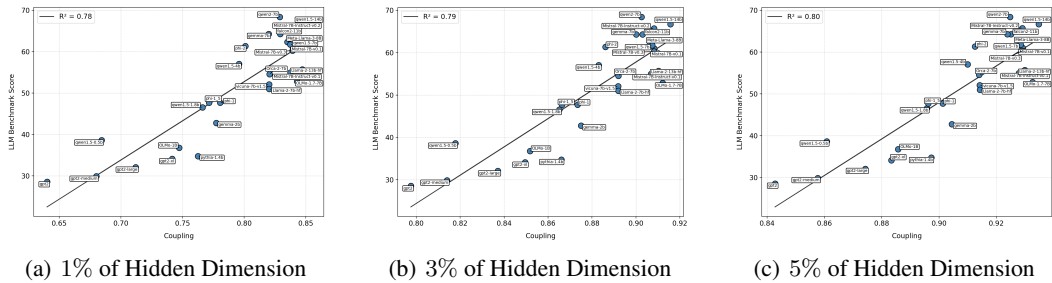

(a) 1% of Hidden Dimension        (b) 3% of Hidden Dimension        (c) 5% of Hidden Dimension

Figure 8: **Coupling plotted against benchmark score for varying percentage of top singular vectors**. This indicates that the relationship is robust across various proportions of top singular vectors.

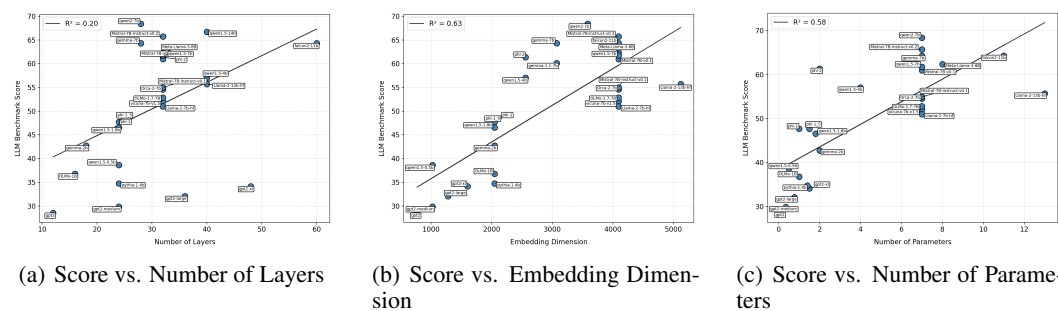

(a) Score vs. Number of Layers        (b) Score vs. Embedding Dimension        (c) Score vs. Number of Parameters

Figure 9: **LLM number of layers, embedding dimension, and number of parameters, against score**.

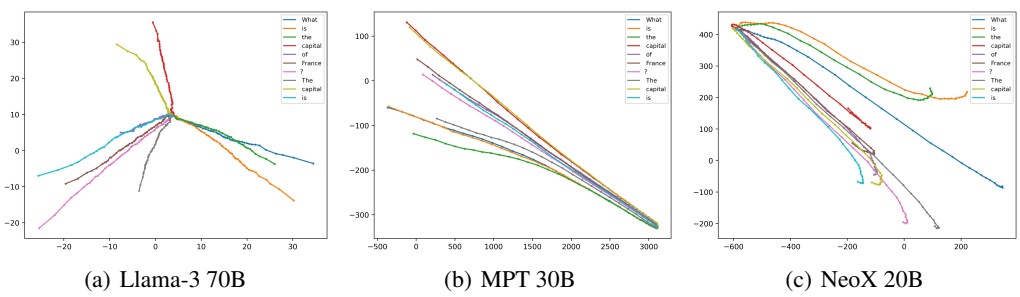

(a) Llama-3 70B        (b) MPT 30B        (c) NeoX 20B

Figure 10: **Trajectories of Hidden Representations.** Visualization of the layer-wise trajectories of hidden representations in Llama 3 70B, MPT 30B, and NeoX 20B in the prompt: `What is the capital of France?  The capital is`. Trajectories of tokens are plotted in latent space, visualized with a 2-dimension principal component projection. A clear directed and outward growth is visible in each token trajectory.

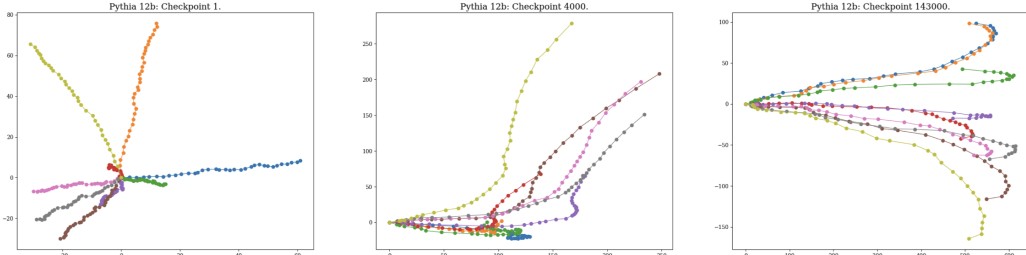

Figure 11: **Evolution of Hidden Trajectories Throughout Training.** Principle component visualizations of the hidden trajectories in Pythia 12B at training checkpoints 1, 4000 and 143000 on the prompt: `What is the capital of France?  The capital is.`

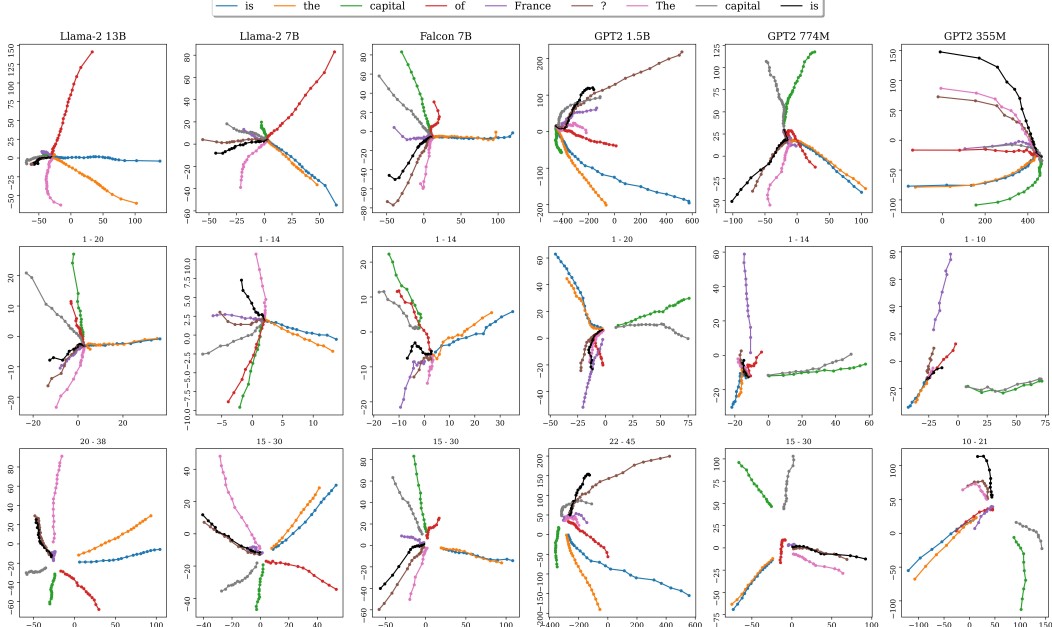

Figure 12: **Hidden Trajectories in LLMs.** Principal components of the trajectories of the hidden representations through various LLMs (columns, decreasing in model size, see Table 1) in the prompt: `What is the capital of France?  The capital is.` Top row: all layers. Middle Row: layers in shallower transformer blocks (layers specified above plot). Bottom Row: layers in deeper transformer blocks (layers specified above plot). Trajectories of each input token (last token 'is' is plotted in black) are plotted in latent space, visualized with a 2-dimension principal component projection. Representations proceed in distinct outward directions, especially in the second half of transformer blocks (lower row) during which the norm of representations increases, with possible abrupt change in the last layer (outer points in upper row). A clear direction of movement is visible in each token trajectory.

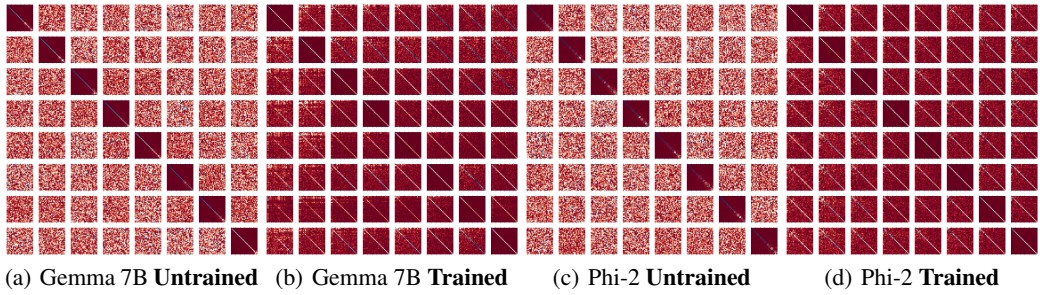

(a) Gemma 7B **Untrained**  (b) Gemma 7B **Trained**  (c) Phi-2 **Untrained**  (d) Phi-2 **Trained**

Figure 13: **Transformer Block Coupling across Depth**.

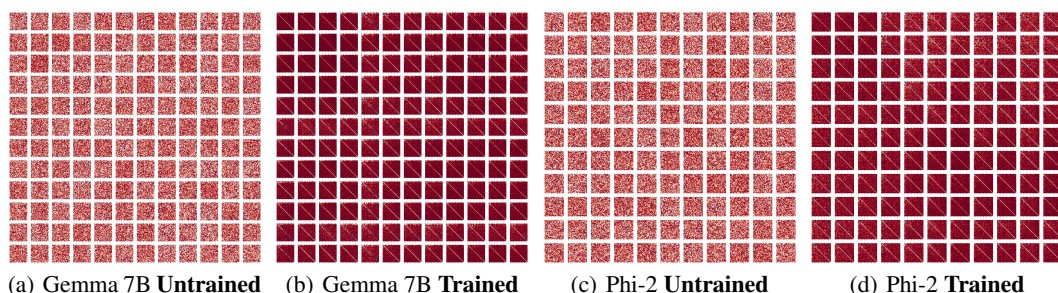

(a) Gemma 7B **Untrained**  (b) Gemma 7B **Trained**  (c) Phi-2 **Untrained**  (d) Phi-2 **Trained**

Figure 14: **Transformer Block Coupling across Tokens (Same input and output tokens)**.

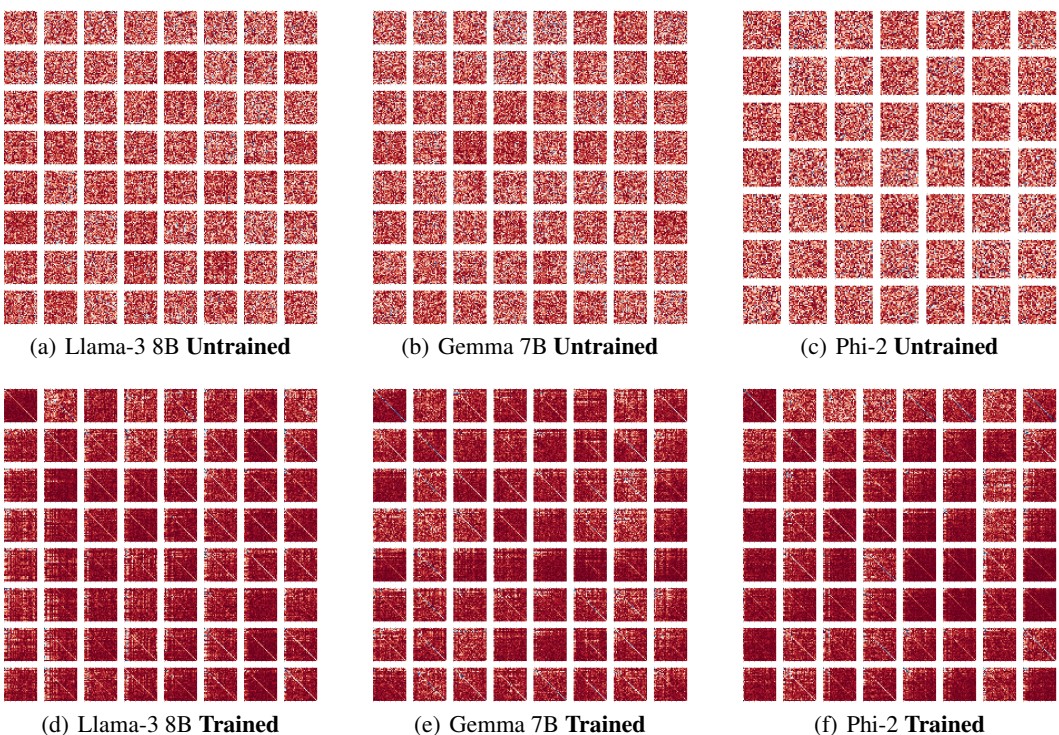

(a) Llama-3 8B **Untrained**          (b) Gemma 7B **Untrained**          (c) Phi-2 **Untrained**

(d) Llama-3 8B **Trained**          (e) Gemma 7B **Trained**          (f) Phi-2 **Trained**

Figure 15: **Transformer Block Coupling across Token (Fixed input)**. The figure illustrates coupling of Jacobians, with fixed input token, across tokens. More specifically, in the matrix plot located at entry $(t_2, t_2')$, the absolute values of the entries of matrices $A_{ll'}^{t_1 t_2 t_1 t_2'}$ are visualized (with randomly fixed layers $l, l'$). In the trained plots (bottom row), the off-diagonal entries being close to 0 with visible diagonal indicates coupling of these Jacobians. This coupling, however, seems to be more evident for certain token pairs and less for others. At initialization (top row), there is no such coupling across tokens. Additional visualizations are included in Appendix A.8 (Figure 19)

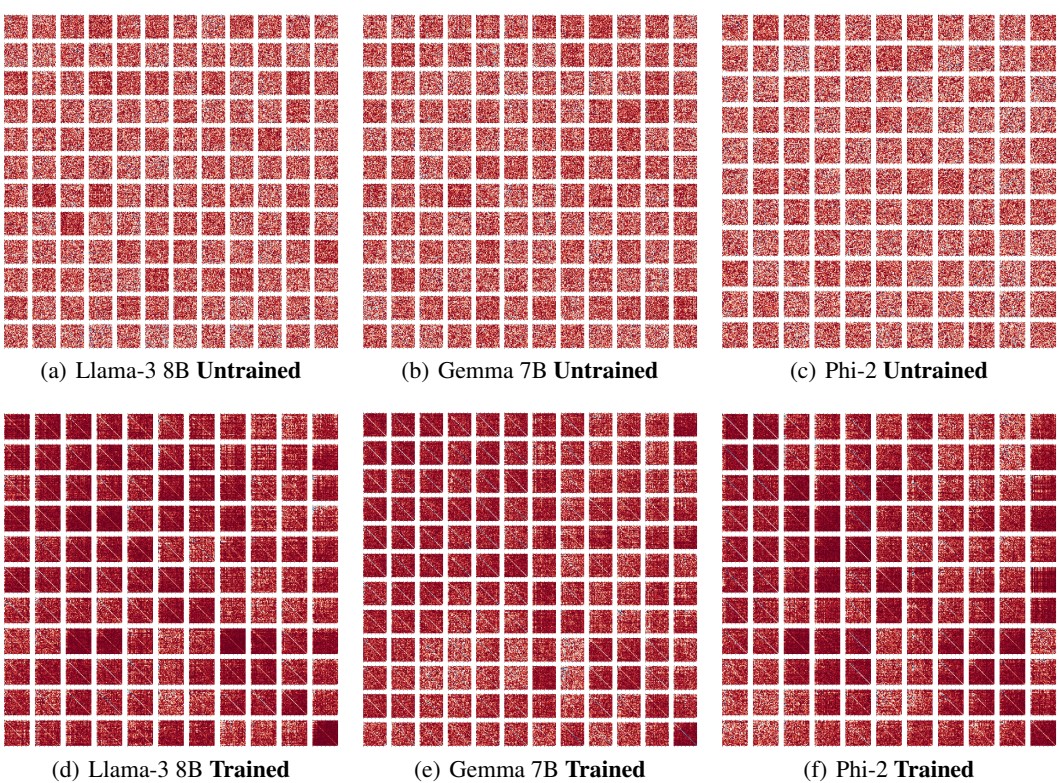

(a) Llama-3 8B **Untrained**          (b) Gemma 7B **Untrained**          (c) Phi-2 **Untrained**

(d) Llama-3 8B **Trained**          (e) Gemma 7B **Trained**          (f) Phi-2 **Trained**

Figure 16: **Transformer Block Coupling across Token (Fixed output)**. The figure illustrates coupling of Jacobians, with fixed output token, across tokens. More specifically, in the matrix plot located at entry $(t_1, t_1')$, the absolute values of the entries of matrices $A_{ll'}^{t_1 t_2 t_1' t_2}$ are visualized (with randomly fixed layers $l, l'$). In the trained plots (bottom row), the off-diagonal entries being close to 0 with visible diagonal indicates coupling of these Jacobians. This coupling again seems to be more evident only for certain token pairs. At initialization (top row), there is no such coupling across tokens. Additional visualizations are included in Appendix A.8 (Figure 20)

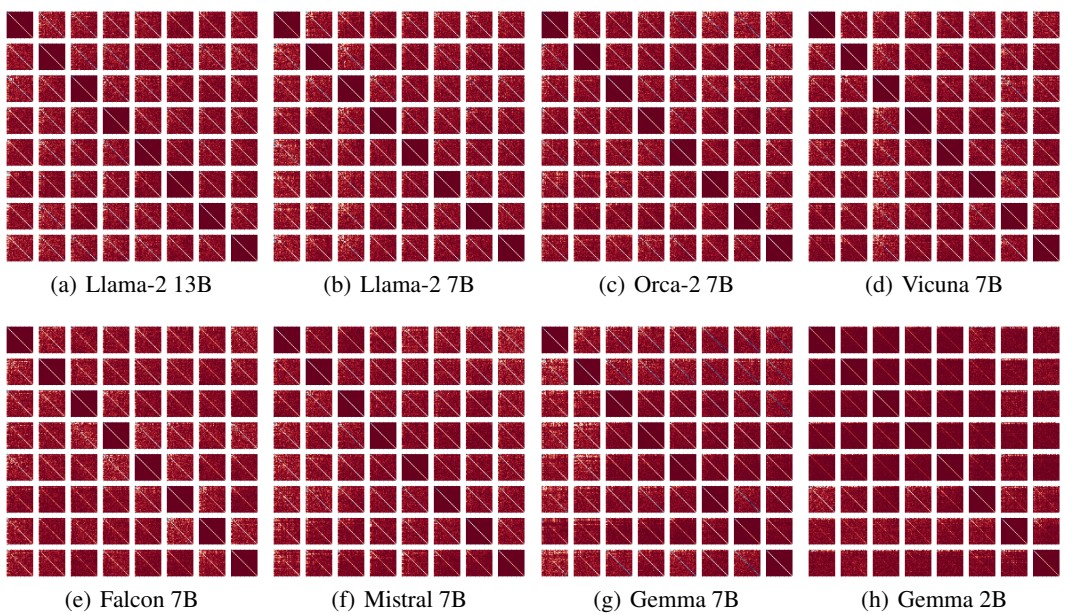

|   |   |   |   |
|---|---|---|---|
| (a) Llama-2 13B | (b) Llama-2 7B | (c) Orca-2 7B | (d) Vicuna 7B |
| (e) Falcon 7B | (f) Mistral 7B | (g) Gemma 7B | (h) Gemma 2B |

Figure 17: **Additional plots of Coupling across depth**. The figure illustrates the coupling of Jacobians across transformer blocks 9 to 16.

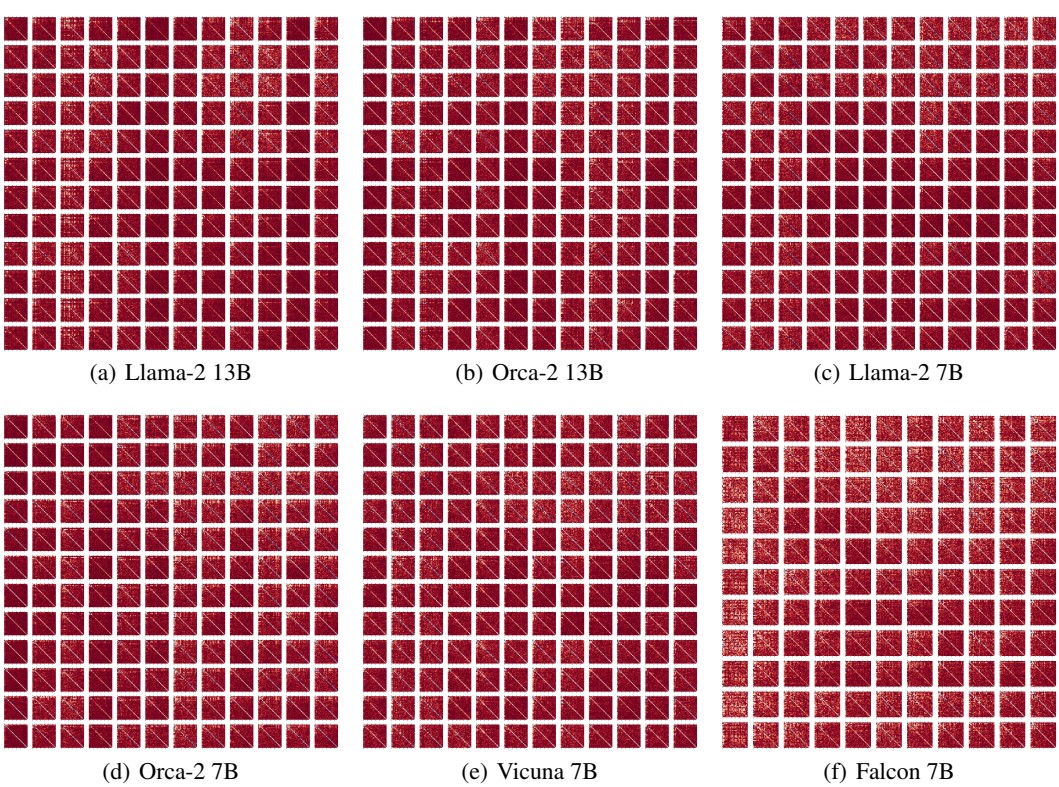

|   |   |   |
|---|---|---|
| (a) Llama-2 13B | (b) Orca-2 13B | (c) Llama-2 7B |
| (d) Orca-2 7B | (e) Vicuna 7B | (f) Falcon 7B |

Figure 18: **Additional plots of Coupling across Tokens (same input and output tokens)**.

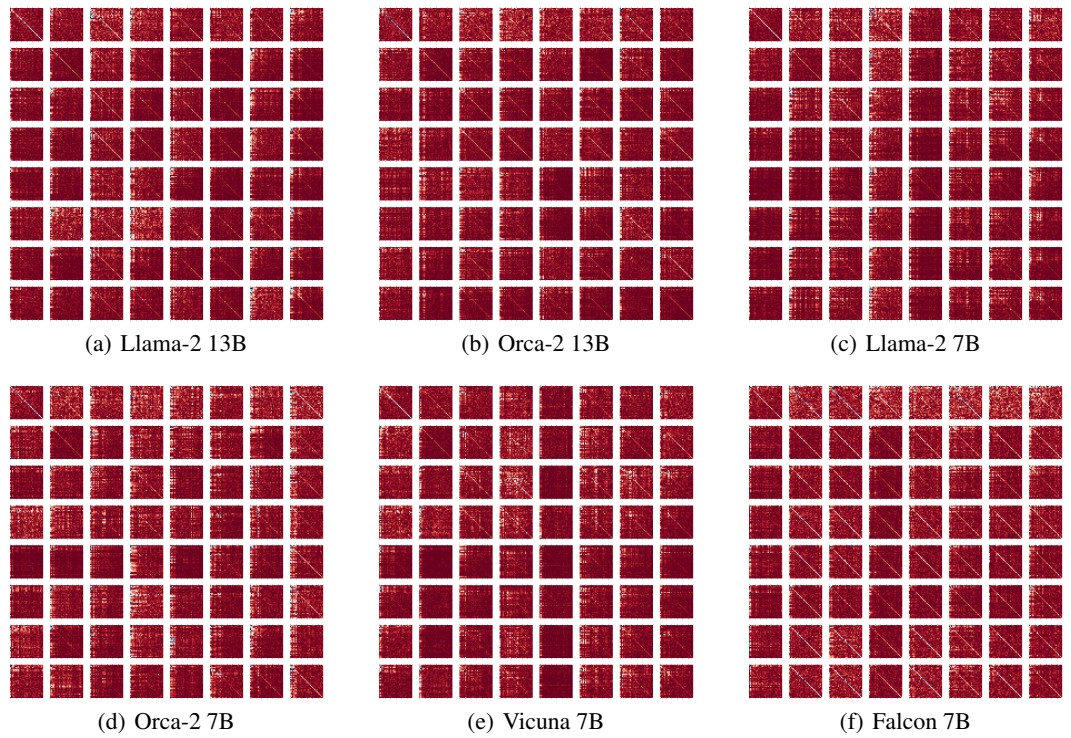

| (a) Llama-2 13B | (b) Orca-2 13B | (c) Llama-2 7B |
| (d) Orca-2 7B | (e) Vicuna 7B | (f) Falcon 7B |

Figure 19: **Additional plots of Coupling across Tokens (fixed input)**.

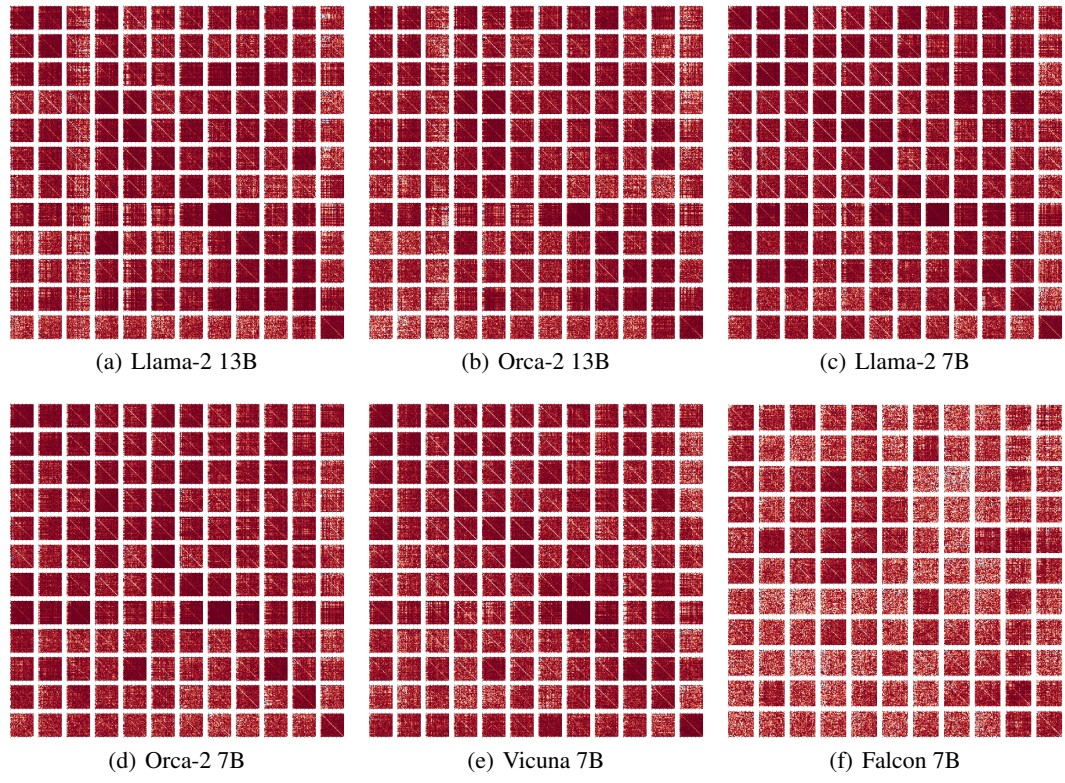

| (a) Llama-2 13B | (b) Orca-2 13B | (c) Llama-2 7B |
| (d) Orca-2 7B | (e) Vicuna 7B | (f) Falcon 7B |

Figure 20: **Additional plots of Coupling across Tokens (fixed output)**.

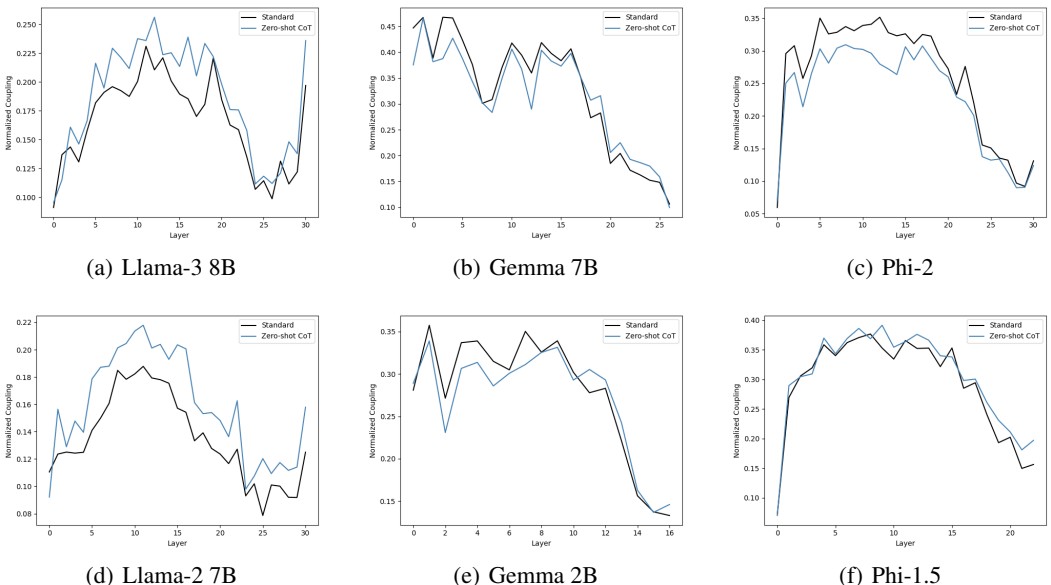

Figure 21: **Zero-shot Chain of Thought on Depth-wise Coupling across Layers**. We compare the normalized coupling on prompts from GSM8k with that of the same prompts appended with "Let's think step by step.", which we refer to as the Zero-Shot CoT (Kojima et al., 2023) prompts. For a more thorough analysis, we measure how much each layer is coupled with all other layers, as shown in the figures above. Firstly, the coupling across layers exhibits distinct behaviors across different models, but with noticeable similarities within the same model families In the LLaMA models, coupling starts off lower, increases in the middle layers, then decreases before showing a slight increase again at the final layers. In the Gemma models, coupling begins relatively high and steadily decreases toward the end of the network. In contrast, the Phi models exhibit significantly lower coupling in the first layer, followed by an immediate increase, and then a slight decrease in coupling toward the final layers. The CoT prompt produces similar coupling patterns to the standard prompt, with slight variations in coupling strength. Specifically, in the LLaMA models, the CoT prompt consistently results in higher coupling across layers. For the Gemma models, the CoT prompt leads to similar overall coupling levels, though some layers exhibit slightly lower coupling and others slightly higher. On the other hand, Phi-2 shows consistently lower coupling with the CoT prompt, while Phi-1.5 is marginally higher. This variability in behavior, along with the similarities within model families, is likely due to differences in training methods and data across organizations, while models within the same family are trained with potentially similar methodologies.

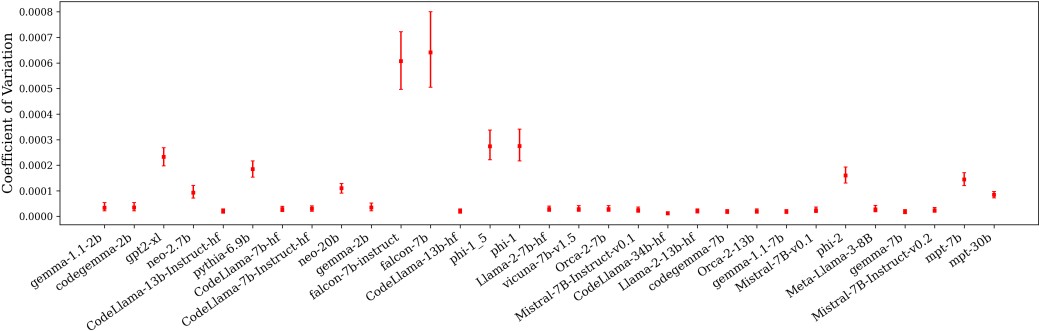

Figure 22: **Coefficient of variation of layer-wise equidistance.** Variation of layer-wise equidistance (Section 3.3) computed over 1,200 prompts from the HuggingFace Open LLM Leaderboard datasets (Section 4.2) on a suite of untrained LLMs (Appendix A.1). Plotted are the median values over all prompts, and are accompanied with uncertainty intervals depicting the inter-quartile range of the results for each model. The models are sorted by increasing benchmark performance.

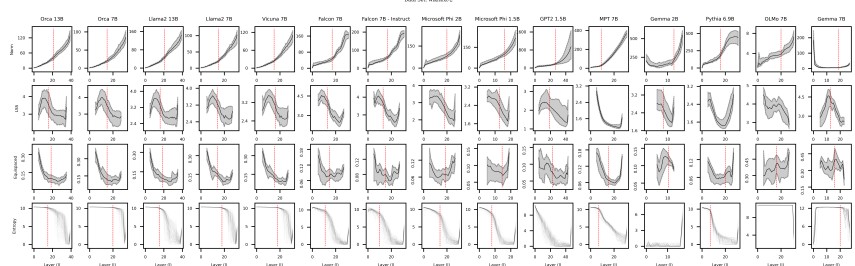

Figure 23: **Various Measurements of Representations.** All measurements were made on 100 prompts taken from the WikiText 2 datasets. (**Row 1**) Norm of hidden representations as a function of layer depth. (**Row 2**) Line Shape Score (LSS) of the hidden trajectories as a function of layer depth. (**Row 3**) Mean equidistance of contiguous hidden trajectories as a function of depth. (**Row 4**) Entropy of logit vectors as a function of depth. Noted in most plots is a line where the behaviour of the measurement drastically changes.

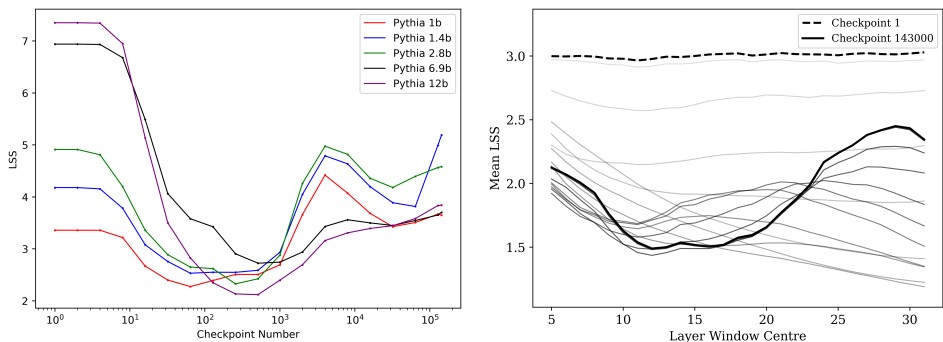

Figure 24: **Linearity Emerges with Training.** Two plots displaying the evolution of the linearity of the token trajectories through training. (**Left**) The LSS as a function of training checkpoint for the variants of the Pythia Scaling Suite Biderman et al. (2023). Here, the LSS is measured over each entire prompt. (**Right**) The mean LSS as a function of layer depth measured at various checkpoints throughout the Pythia 12B model. Here, the LSS is computed on a window of layers of width 11, centred at the value given by the x-axis.

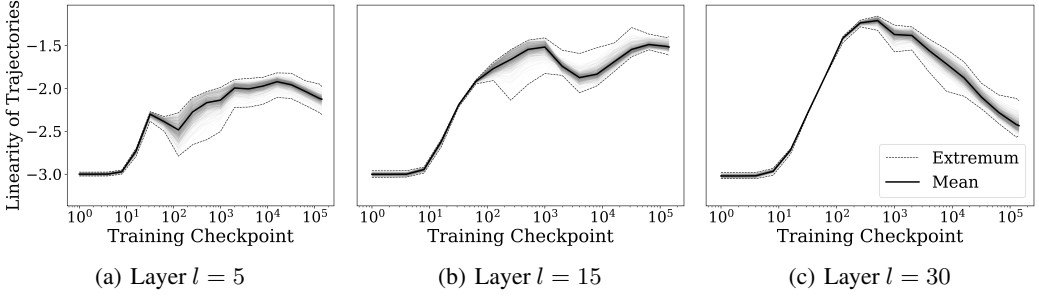

Figure 25: **Emergence of Linearity with Training.** Average linearity of a trajectory at block depths $l \in \{5, 15, 30\}$ evaluated for Pythia 12B (Biderman et al., 2023) checkpoints $\{1, 2, 4, \ldots, 256, 512, 1k, 2k, 4k, \ldots, 128k, 143k\}$. The linearity is given by the negative LSS, and is computed on a window of 11 layers centered at each depth $l$.

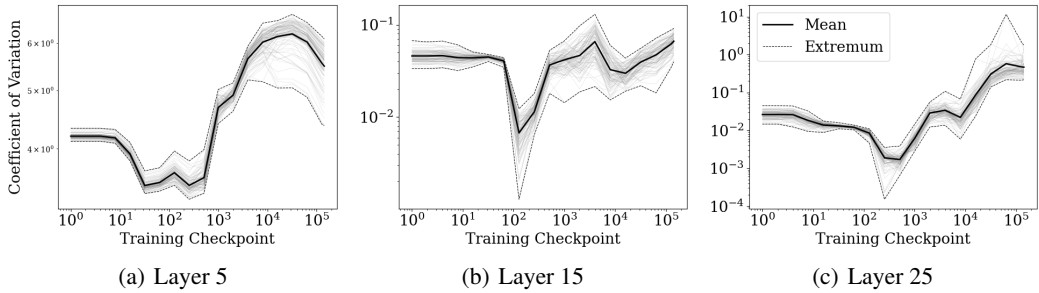

(a) Layer 5         (b) Layer 15         (c) Layer 25

Figure 26: **Expodistance at Fixed Layers.** Plotted are mean expodistances as a function of training checkpoint at various depths of the network. The values at a given depth are the mean expodistance over a layer window of width 11 centred at said depth 100 MMLU prompts are plotted at each layer.

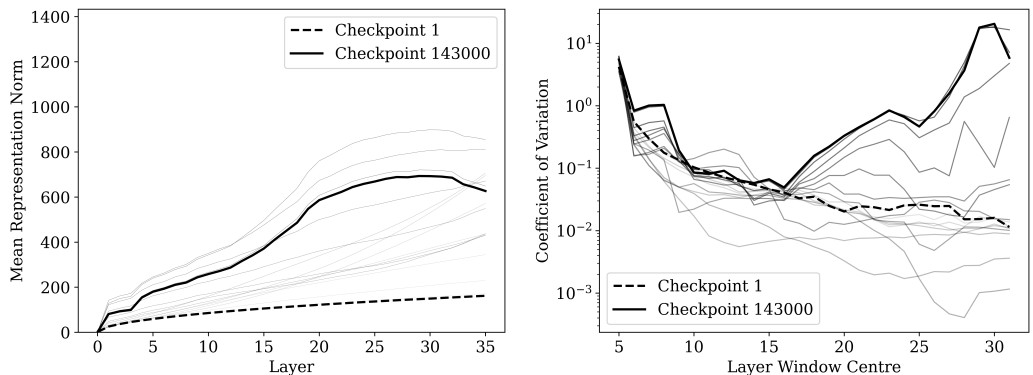

Figure 27: **Norm and Expodistance During Training.** (**Left**) Plotted is the norm of the representations as a function of depth at various training checkpoints. Observed is the transition form log-like growth in early stages to exponential-like growth, particularly through layers 5 through 20, as training evolves. (**Right**) Plotted is the expodistance over a layer window of width 11 centred at the give depth, each computed at a variety of training checkpoints.

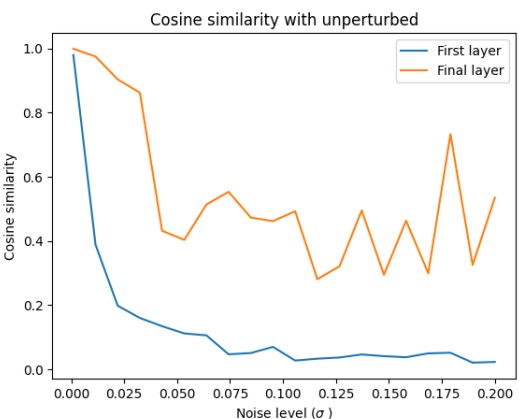

Figure 28: **Perturbation Experiments with Llama-3 8B.** The last token embedding is perturbed with various noise levels, and compared with the true embedding at the first and last layers. The trend shows that at small noise levels, cosine similarity with the true embedding remains somewhat high, and is significantly lower at the first layer.

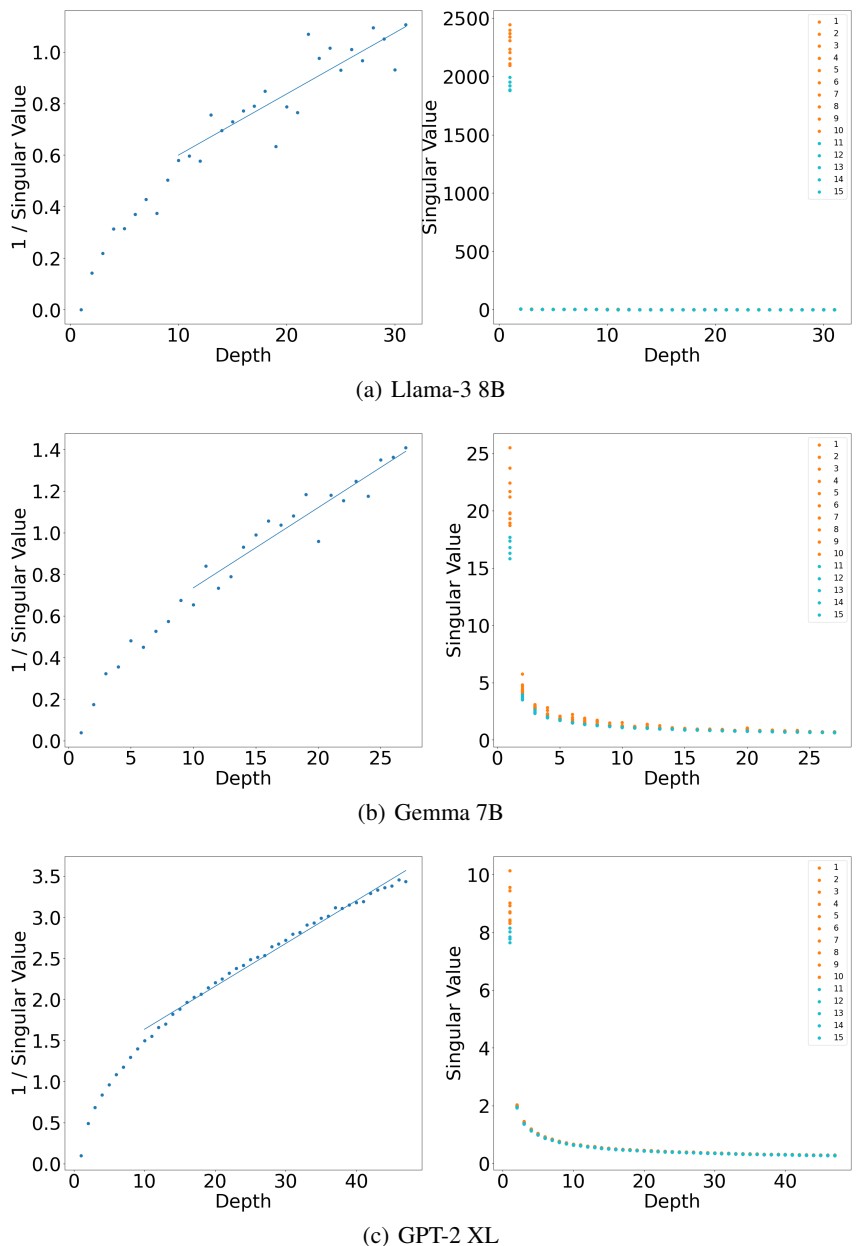

(a) Llama-3 8B

(b) Gemma 7B

(c) GPT-2 XL

Figure 29: **Scaling of Singular Values of Residual Jacobians (Untrained).**

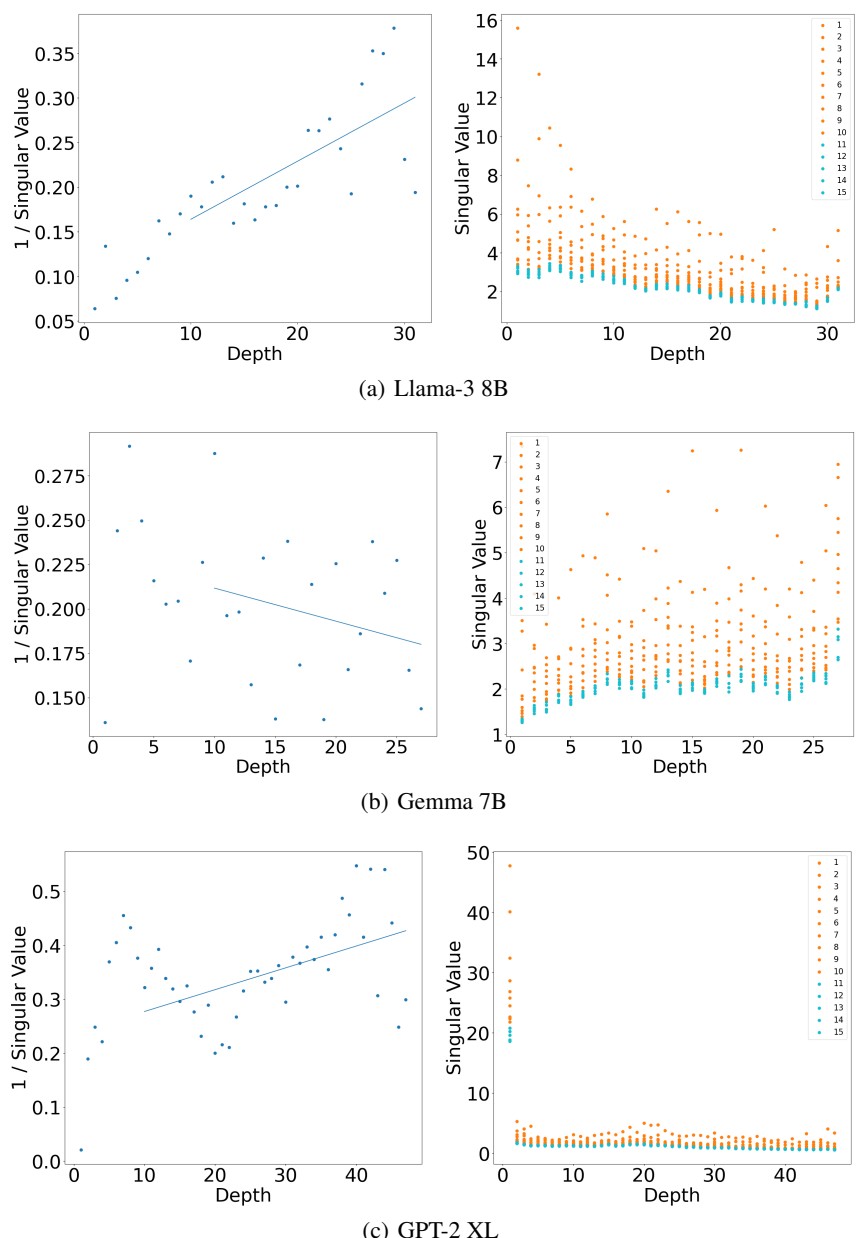

(a) Llama-3 8B

(b) Gemma 7B

(c) GPT-2 XL

Figure 30: **Scaling of Singular Values of Residual Jacobians (Trained)**.

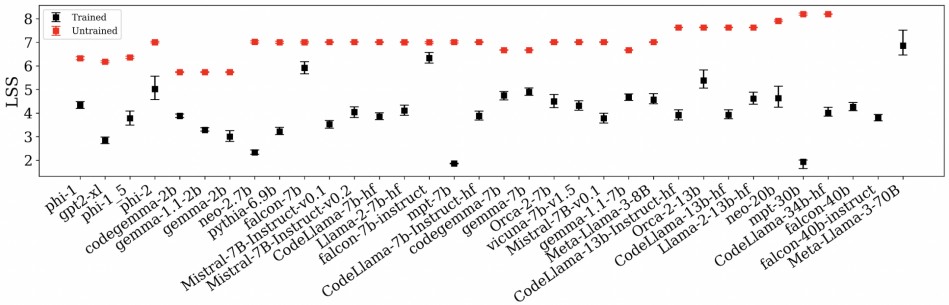

Figure 31: **LSS sorted by LLM parameters**.

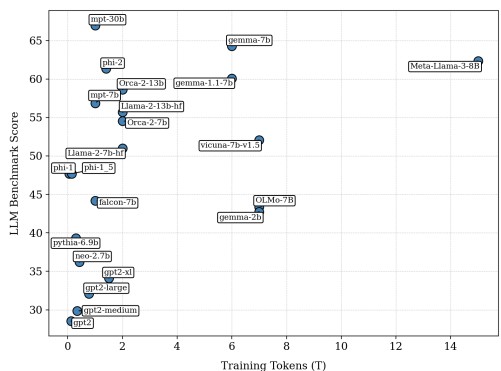

Figure 32: Plotting the number of training tokens against its LLM Huggingface Benchmark score.

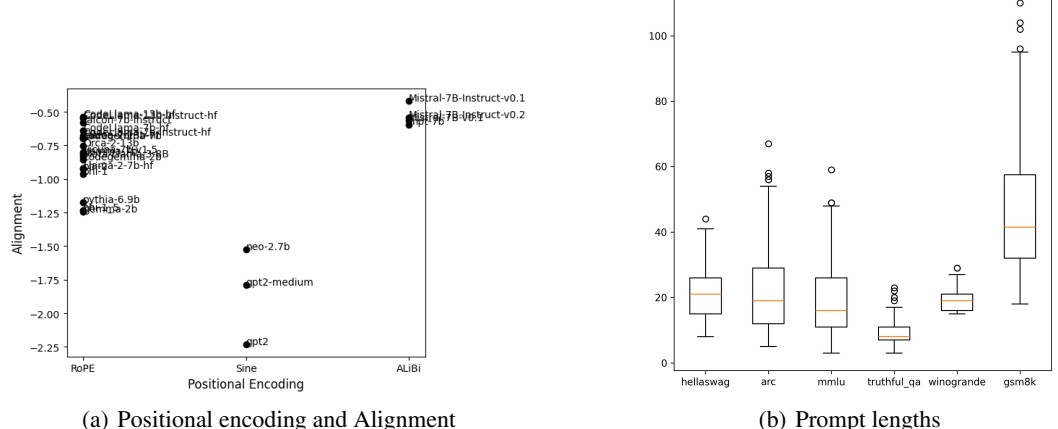

(a) Positional encoding and Alignment

(b) Prompt lengths

Figure 33: **Other plots**.

