# OpenReview forum: "Transformer Block Coupling and its Correlation with Generalization in LLMs"
_ICLR.cc/2025/Conference — ICLR 2025 Poster_

### Official Review · Reviewer_8ECx · 2024-10-19

**Soundness:** 3
**Presentation:** 2
**Contribution:** 2
**Rating:** 5
**Confidence:** 4

**Summary:**

The paper proposes Coupling, a metric of a Transformer model.
It computes the similarity of two Jacobians, followed by a normalization; each is the sum of non-residual terms by Attention and FFN.
The main claim of this paper is that the Couping metric of a model, among a set of public pre-trained LLMs, is correlated with the average performance of a set of downstream tasks from the HuggingFace Open LLM Leaderboard (R^2 value of 0.7214).
Additionally, the paper explores the properties of the Transformer models in terms of the line-shape score (LSS), defined by Gai & Zhang (2021), and the exponential spacing (expodistance) of the hidden trajectories.

**Strengths:**

To the best of my knowledge, this work is the first to explore the Coupling measures and indicate a correlation between them and the LLM performance on downstream tasks.
It also analyzes linearity in intermediate embeddings over the depth and exponential growth of the embedding norms.

**Weaknesses:**

The correlation between the Coupling metric and performance may not be a helpful or exciting finding. First of all, the paper does not provide any reasonable explanation or hypothesis about (1) why they investigated the Coupling, (2) why the Coupling could reflect performance, and (3) whether the correlation is from a causality (of a direction) or not.
Additionally, the paper does not perform intervention experiments, e.g., training a model with a regularization of increasing the Coupling and demonstrating the improvement. In total, it is difficult to conclude that this finding is helpful for the community so far. There are many experimental results, but they are not always well-organized and explained. I would like to know the real contributions of the results.

The correlation between the Coupling metric and performance might be very consistent. It may disappear in trends of recent high-performance models.

The contribution of the analysis of linearity and exponential growth is unclear. In my understanding, the analysis is conducted independently from the Coupling. Results are not specific to the models of interest and the Coupling. Many things might be obvious from Li & Papyan (2023). It would be great to add connections between them and Coupling (and the performance, if possible).

**Questions:**

In summary, what can we concretely learn from the result? If we don't consider the quality, it's not very difficult to develop metrics that show a correlation but are ultimately not useful. Playing devil's advocate for a moment, for example, one of the most boring metrics could be the performance of a single downstream task like GSM8K. It may be correlated with the performance on the LLM Leaderboard. But, of course, this finding is not helpful. Compared with this or other candidates with such correlations, how can we claim that the Coupling metric is more interesting, helpful, or convincing?

The main claim may be based on Figure 1. But, the result looks very consolidated. Can we believe the presence of the correlation over various models or a limited number of models? For example, what happens if you only examine models that score above 50? Is there still a strong correlation, or does the relationship break down at higher performance levels? For example, if one observes phi-1 and phi-2, the Coupling rule may be broken because they may have similar settings, similar Couplings, and very different performances.

I'm not sure why linearity and exponential growth are discussed and considered as a goodness metric. Playing devil's advocate for a moment again, if we make Transformer blocks, which just scale their inputs by large positive constants, it seems to satisfy the linearity and exponential growth but would fail to achieve the high performance as they the Transformer is just a scaler. Could you provide more discussion and reasons for choosing or designing the metrics for analysis?

Figure 3 (c). Does this show that Linearity disappears at the end of training?

Figure 6. MPT -> Mistral?

Can we see Figure 6-style results of untrained models?

Figure 8. Can we see the results of untrained models?

---

> ### Author Response · Authors · 2024-11-22
>
> We thank the reviewer for the thorough review of our manuscript. We appreciate the numerous comments and the opportunity to further clarify several points in our work. Figures 1-3 are clarified to be interpretable, Section 6.2 has major revisions in response to reviewer comments. Additionally, we have trained 64 ViT with varying stochastic depth and weight decay settings to analyze the emergence of coupling in transformers. Figure 7b is now updated in response to the reviewer’s last question.
>
> **“The paper does not provide any reasonable explanation or hypothesis about:
> (1) why they investigated the Coupling.”**
>
> To clarify our motivation, we have added further clarificaiton to the introduction of the main text
>
> > In our work, we investigate whether there are identifiable structural characteristics across 38+ pretrained LLMs, measure their emergence with training, and analyze their relationship with generalization performance. During inference, as token embeddings pass through the network, we linearize
> the effect of transformer blocks on the token embeddings throughout the depth of the LLM. To this
> end, we compute the Jacobians of distinct connections between layers or tokens, derive their singular
> value decompositions (SVDs), and compare the resulting singular vectors. This approach measures
> the degree of coupling between singular vectors to capture the operational similarity of blocks as
> they act on tokens.
>
> Additionally, previous work [1] has explored some form of coupling in ResNets. A natural extension of this question is to investigate whether a similar phenomenon occurs in LLMs, which are transformer-based and several orders of magnitude larger, and to determine if any of these properties are related to performance. Our study addresses this gap, revealing insights about the internal dynamics of LLMs and in newly included experiments with ViTs.
>
> [1] Li et al. Residual Alignment: Uncovering the Mechanisms of Residual Networks (2024).
>
> **“(2) why the Coupling could reflect performance”**
>
> In response, we have added the following explanation to Section 5.4 in the revised manuscript:
>
> >If the blocks are strongly coupled, they guide representations in consistent directions, tending towards streamlined paths throughout the network. We hypothesize these simpler trajectories may lead to better generalization. This agrees with many generalization bounds in machine learning [2], which suggest that models with lower complexity tend towards better generalization.  Additionally, prior works [3] demonstrate that the Frobenius norm of input-output Jacobians is related to generalization, providing evidence that coupling — a structural property derived from Jacobians — may also correlate with generalization.
>
>
> [2] Arora et al. Stronger generalization bounds for deep nets via a compression approach. (2018)
>
> [3] Novak et al. Sensitivity and Generalization in Neural Networks: an Empirical Study. (2018).
>
> **“(3) whether the correlation is from a causality (of a direction) or not.”**
>
> In the present work, the coupling is correlated with generalization, and we do not claim the presence of causality. One way to demonstrate a causal relation would be through a theoretical derivation which might prove that coupling occurs when the model generalizes, and would rely on the empirical foundations laid in our work.
>
> The reviewer comment prompted further investigation of coupling in small ViTs. Figure 6 now demonstrates the presence of coupling in 64 ViT classifiers on CIFAR10, with varied weight decay and stochastic depth. We observe that coupling correlates with test accuracy when each stochastic depth rate is fixed (Figure 6a). In addition, stochastic depth improves coupling (Figure 6b) for fixed weight decay values. We hypothesize that stochastic depth causes transformer blocks to be operationally similar and consequently encourages coupling.

---

> ### Author Response · Authors · 2024-11-22
>
> **“It is difficult to conclude that this finding is helpful for the community so far… I would like to know the real contributions of the results.”**
>
> Our contribution establishes a fundamentally new phenomenon in transformers across many models, which were designed and trained independently by 7 distinct organizations, all featuring the coupling phenomenon and regularity within their internal representations. We are not aware of such a measurement that consistently emerges within all measured open LLMs. Transformer Block Coupling is helpful to the community for the following reasons:
>
> 1. Previous works [5] have successfully pruned LLMs by skipping transformer blocks in the forward pass. Our work suggests that many transformer blocks are operationally similar, with significant regularity in the hidden trajectories that has previously been overlooked, which supports past works on pruning.
> 2. A related work to this is on Neural Collapse and its occurrence in LLMs [6]. Our work shows that the mechanism that leads to NC may be related to transformer block coupling.
> 3. Past work [7] provides mathematical derivations of the dynamics of transformers under the assumption that all layers are coupled. Our work provides empirical evidence for part of their perspective.
> 4. The ALBERT transformer [8] proposes applying the same transformer block weights to each depth. Our work provides empirical evidence that the blocks are operationally similar once linearized through Jacobian matrices.
> 5. ViT measurements show that stochastic depth improves coupling and is known to improve generalization. To a certain extent, they shed light on why these techniques work in practice. Our findings suggest that coupling may provide new insight into the underlying mechanism of stochastic depth and related techniques.
> 6. As demonstrated by our findings in stochastic depth amplifying coupling (Figure 4b),  developing training methods to promote coupling across transformer locks could provide additional regularization and improve performance.
>
> We have revised the “related works” section to further discuss many of these applications and implications.
>
> [5] Gromov et al. The Unreasonable Ineffectiveness of the Deeper Layers (2024).
>
> [6] Wu et al. Linguistic Collapse: Neural Collapse in (Large) Language Models (2024).
>
> [7] Geshkovski et al. The emergence of clusters in self-attention dynamics (2024).
>
> [8] Lan et al. ALBERT: A Lite BERT for Self-supervised Learning of Language Representations (2019).
>
> **“The contribution of the analysis of linearity and exponential growth is unclear. In my understanding, the analysis is conducted independently from the Coupling.”, “Many things might be obvious from Li & Papyan (2023).”**
>
> We have extended the discussion in Section 6.1 with the following to explicitly relate the observed empirical trends of exponential growth and linearity to the theoretical perspective outlined below.
>
> > Suppose that the representations follow the linearized equation:
> $$ x^{l+1} = x^l + J_l x^l = (I + J_l)x^l$$
> Expanding across layers, the entire system can be approximated by the product
> $$ x^L = (I + J_L)(I + J_{L-1})\cdots (I + J_1)x^0 $$
> Assuming that $J_l \approx USU^T$, this equation predicts that the norm of $ x_l $ would exhibit exponential growth layer by layer. Expanding $U$ and $S$,
> $$ x^l = \sum_{j=1}^{d_\text{model}} u_j (1+s_j)^l u_j^\top x^0 $$
> where $u_j $and $ s_j$ represent the eigenvectors and eigenvalues of the Jacobian, respectively. The above equation predicts that the norm of $ x^l $ would exhibit exponential growth layer by layer. In general, trajectories are not expected to be perfectly linear unless $ x^0 $ aligns with an eigenvector of $ J $. However, in our experiments, we observe a notable tendency towards linearity, suggesting that the representations align progressively during training with the eigenvectors of the coupled Jacobians.
> Exponential growth is a distinct characteristic of transformers, since in ResNets it was observed that trajectories are equispaced [9].
>
> We now include additional details for the choice of metrics in Appendix A4.
>
> [9] Li et al. Residual Alignment: Uncovering the Mechanisms of Residual Networks (2024).

---

> ### Author Response · Authors · 2024-11-22
>
> **“It's not very difficult to develop metrics that show a correlation but are ultimately not useful.”**
>
> We thank the reviewer for raising this point. Our metrics are designed to reflect intrinsic properties of the models, rather than merely external trends. The observed correlation between coupling and performance is a relationship between an internal property of the model and its performance, rather than a purely incidental association.
>
> In response, we now include additional comparison between the correlation of benchmark scores and other relevant hyperparameters (Figure (8b)). These comparisons highlight that the observed correlation for our metric is significantly stronger.
>
> Our metrics are motivated by the discussion in Section 6 and previously mentioned theoretical works. The coupling measurements relate the operation of transformer blocks to their fundamental decompositions, and are correlated with generalization. Our measurements provide insight that underscores past theoretical investigations by the community, in particular, for demonstrating empirical evidence in LLMs and ViTs for some transformer dynamics perspectives [10].
>
> [10] Geshkovski et al. The emergence of clusters in self-attention dynamics (2024).
>
> **“Compared with this or other candidates with such correlations, how can we claim that the Coupling metric is more interesting, helpful, or convincing?”**
>
> We appreciate the reviewer’s point of clarification. The key distinction of the coupling metric lies in its nature as a structural property of the representations, and therefore a property of the model itself, rather than an arbitrary metric designed solely to correlate with performance.
>
> Coupling provides meaningful insights into the underlying mechanisms of the model, specifically the coordination and operational similarity between transformer blocks. This stands in contrast to metrics that merely measure task performance or are derived from outputs without offering an explanation of the internal model dynamics. By examining coupling, we gain an operational understanding of how different parts of the model interact and contribute to its overall behavior, making it both more interesting and useful for advancing our understanding of transformers.
>
> **“Can we believe the presence of the correlation over various models or a limited number of models?”, “For example, what happens if you only examine models that score above 50?”.**
>
> We appreciate the question, and address it below:
> 1. **Breadth of Evaluation.** The coupling measurement has been conducted across numerous open-source models available on HuggingFace. Notably, this analysis is novel, as coupling had not been observed in transformers prior to our results. Our study demonstrates a significant correlation between coupling and performance across a large sample of models, which supports the generality of the finding.
> 2. **Updated Analysis on subset of models.** In the revised manuscript, we have included a new coupling-versus-generalization plot (Figure 1a) that was generated using a larger set of prompts. To directly address the reviewer’s question, we also present a separate plot restricting to models with higher scores, as shown in Figure (39). While the correlation in this subset is weaker than the global trend, it remains nontrivial, reinforcing the validity of our observations even within this constrained range.
> 3. **Task-Specific Experimentation.** To further strengthen the analysis, we conducted an additional experiment examining block coupling across various task types. By measuring coupling as a function of performance on specific tasks, we observed positive linear correlations, which suggest the robustness of coupling as a meaningful metric. These task-specific findings are detailed in Table 2 in the appendix.
>
> We believe these updates comprehensively address concerns regarding the reliability and scope of the observed correlation.

---

> ### Author Response · Authors · 2024-11-22
>
> **“I'm not sure why linearity and exponential growth are discussed and considered as a goodness metric.”, “If we make Transformer blocks, which just scale their inputs by large positive constants, it seems to satisfy the linearity and exponential growth but would fail to achieve the high performance”**
>
> We thank the reviewer for the point of clarification. In response, the following discussion is now included in Section 6.2.
> >Coupling suggests that the representations follow the linearized equation:
> $$ x^{l+1} = x^l + J_l x^l = (I + J_l)x^l $$
> Expanding across layers, the entire system can be approximated by the product
> $$ x^L = (I + J_L)(I + J_{L-1})\cdots (I + J_1)x^0  $$
> Assuming that $J_l \approx USU^T$, this equation predicts that the norm of \( x_l \) would exhibit exponential growth layer by layer. Expanding $U$ and $S$,
> $$
>    x^l = \sum_{j=1}^{d_\text{model}} u_j (1+s_j)^l u_j^\top x^0
> $$
> where  $ u_j $ and $ s_j $ represent the eigenvectors and eigenvalues of the Jacobian, respectively. The above equation predicts that the norm of $ x^l $ would exhibit exponential growth layer by layer. In general, trajectories are not expected to be perfectly linear unless $ x^0 $ aligns with an eigenvector of $ J$. However, in our experiments, we observe a notable tendency towards linearity, suggesting that the representations align progressively during training with the eigenvectors of the coupled Jacobians.
>
> Linearity and exponential growth are not intended as direct performance metrics, but rather they are structural properties observed in the model's trajectories. These properties help characterize how inputs evolve through the model, and are noteworthy because they occur while the model still maintains high performance. As the reviewer highlights, blocks that simply scale inputs by large constants may exhibit linear trajectories and exponential growth, but such transformations would not lead to meaningful outputs or good performance. The key distinction is that LLMs **tend towards** these properties alongside effective learning and performance.
>
> **Could you provide more discussion and reasons for choosing or designing the metrics for analysis?**
>
> Appendix A4 is now updated with a discrete formulation, with enhanced discussion of the relationship to our metrics. Our motivation to measure the linearity of the trajectories is drawn from [11], [12], and to quantify the properties of visualizations in Figures (12, 13, 14). Linearity results in LLMs are in analogy with those previously observed in ResNets. Exponential growth distinctly emerges in transformers, in contrast to ResNets, in which equispaced trajectories are observed. This observation is justified in our discussion on Section 6.2 below.
>
> [11] Gai et al. A Mathematical Principle of Deep Learning: Learn the Geodesic Curve in the Wasserstein Space (2021)
>
> [12] Li et al. Residual Alignment: Uncovering the Mechanisms of Residual Networks (2024).
>
> **Figure 3 (c). Does this show that Linearity disappears at the end of training?**
>
> According to our results, as displayed in Figure 3(c) (now Figure 28c), the linearity of trajectories tends to decrease slightly in the deeper layers of the network by the end of training, while still remaining more linear than earlier in training. This does not suggest that the trajectories become entirely unstructured.
>
> Overall, LSS is a measure of whether the trajectories exhibit regularity, specifically in terms of their tendency towards linearity. As shown in Figure 3, the degree of linearity varies with depth and evolves throughout training. This variation highlights the nuanced behavior of trajectories across layers and different training stages, rather than indicating a complete disappearance of linearity at any point.
>
> **Figure 6. MPT -> Mistral?**
>
> MPT is a model by Mosaic ML, and is cited as such.
>
> **Can we see Figure 6-style results of untrained models?**
>
> A PCA of the trajectories for the untrained models appears in Figure 13 of the updated document.
>
> **Figure 8. Can we see the results of untrained models?**
>
> In response to the question, we have measured exponential growth of representations in the untrained models. Figure 7b of the updated manuscript now contains both the trained and untrained exponential distancing results, with trained models being consistently more exponentially spaced than the untrained ones. We thank the reviewer for this suggestion since it led to notable increases in the depth of our analyses.
>
> In closing, we'd like to express again our gratitude to the reviewer for their feedback which not only improved the current state of our manuscript but also set the stage for intriguing future investigations. In light of these refinements, should the reviewer find it fitting, we would be most grateful for any potential increase in our score.

---

> > ### Comment · Reviewer_8ECx · 2024-11-26
> >
> > Thank you for your detailed response and for conducting extensive additional experiments. I acknowledge the substantial effort behind this work. I want to begin by noting that, due to the overwhelming volume of new experiments and analyses, as well as ongoing discussions with other reviewers, I may not yet have fully processed all the new information you have provided. I will take more time to reflect on this as needed.
> >
> > For now, I remain unconvinced about the practical value of the coupling metric and the degree to which the findings are fundamentally novel or non-trivial. Demonstrating a correlation—whether intrinsic or extrinsic—does not automatically make the result "useful" unless it leads to actionable insights. I understand that the manuscript does not claim causality. Then, how exactly this result is beneficial is still a concern.
> >
> > I understand your position that the contribution lies in presenting coupling as a scientifically interesting observation rather than an immediately practical tool. However, the novelty and non-obviousness of these findings remain unclear to me, especially in light of existing Transformer research. For example, while your findings align with prior work, this also raises concerns that they may have been expected based on previous studies. Even if the complexities of your coupling metric were not explicitly anticipated, the qualitative nature of the findings may not represent a truly surprising insight.
> >
> > Looking to the future, I see the potential for this kind of work to inspire further research. However, relying solely on potential future utility is not a strong argument in itself. Without practical applications or a compelling framework demonstrated in this paper, the concerns about its utility remain inevitably. Furthermore, the possible existence of numerous complex metrics that monotonically increase or decrease during training leaves room for skepticism. It is difficult to argue that coupling is uniquely explanatory or consistently meaningful across diverse Transformer or LLM behaviors.
> >
> > Your claim that coupling reveals a "fundamentally new phenomenon in Transformers across many models" also raises questions. As the new Figure 39 shows, the correlation between coupling and performance diminishes significantly among modern high-performing LLMs. This suggests that coupling may be less robust as a comparative metric across different models or training setups. Instead, it risks being a proxy for secondary factors (e.g., training compute or other unknown factors) rather than capturing an essential underlying mechanism.
> >
> > In summary, while I maintain reservations about the scientific significance and practical utility of the findings in this study, I deeply appreciate the thoroughness of your experiments and, especially, the transparent reporting of results. If other reviewers are confident of finding substantial value in these contributions, I will not object to the paper being accepted. Additionally, I commend the authors’ rigorous and honest approach, including reporting results that do not always strengthen their main claims  (like Figure 39).
> >
> > With these considerations, I am raising my score from 3 to 5.
> >
> > -----
> >
> > Minor:
> >
> > Regarding the caption for Figure 6: While the authors clarify that MPT is the correct model, the caption mistakenly refers to "Mistral 30B." Please address this inconsistency.

---

> ### Author Response · Authors · 2024-11-27
>
> We would first like to sincerely thank the reviewer for the continued dedication to analyzing our work. We have addressed each to the best of our abilities below.
>
> **“the qualitative nature of the findings may not represent a truly surprising insight.”
> “...the degree to which the findings are fundamentally novel or non-trivial.”
> “your findings align with prior work”
> “Your claim that coupling reveals a "fundamentally new phenomenon in Transformers across many models" also raises questions.”**
>
> Thank you for your feedback. To the best of our knowledge, our study is the first to systematically investigate the coupling of distinct transformer blocks. If the reviewer views the findings as trivial or obvious based on past results, we would greatly appreciate a citation of prior work that has demonstrated similar measurements or explains the coupling phenomenon. This would help us ensure proper contextualization of our work and aid us in refining our manuscript and better addressing potential gaps.
>
> We believe [1] to be the work which most closely aligns with the efforts of this paper. Our work extends this line of research in several significant ways:
>
> 1. **Quantifying Coupling Strength (Equation 13 and Figure 2):**
> We introduce quantitative metrics, which were not proposed in prior work (which showed only qualitative figures), that quantify the strength and coupling between transformer blocks, enabling precise measurement of pairwise interactions and facilitating comparisons across models.
> 2. **Correlation with Performance (Figure 1a):**
> We investigate the relationship between coupling and model performance, reporting statistically significant correlation between coupling and generalization. Nothing of this sort has been presented in any past work.
> 3. **Training Dynamics (Figures 1[b, c, d, e], 3, 4):**
> We examine how coupling evolves over the course of training, revealing a gradual increase and highlighting its dynamic nature. Nothing of this sort has been presented in prior work.
> 4. **Beyond depth – Token-wise Coupling (Figure 2, 4):**
> Our approach generalizes depth-wise coupling to token-wise coupling which was not studied, even qualitatively, in prior work.
> 5. **Quantifying Trajectory Growth (Figure 5b, Equation 18):**
> We demonstrate, with rigorous analysis across numerous models, the consistent exponential growth of trajectories—a phenomenon not previously quantified, and differing from ResNets.
> 6. **Diversity of Models and Training Contexts (Table 1):**
> We analyze a wide range of open-source models, with billions of parameters, from 7 independent organizations, enhancing the robustness and generality of our findings, and going beyond the residual alignment work which studied ResNets trained on CIFAR.
>
> [1] Li et. al. "Residual Alignment: Uncovering the Mechanisms of Residual Networks". 2024.

---

> ### Author Response · Authors · 2024-11-27
>
> **“I remain unconvinced about the practical value of the coupling metric”, “how exactly this result is beneficial is still a concern.”**
>
> We thank the reviewer for the additional feedback. Our work is positioned within the interpretability and explainable AI track, with the primary goal of providing theoretical and empirical grounding for understanding phenomena in LLMs rather than proposing direct practical improvements.
>
> To clarify the practical value of the coupling metric, we emphasize that our results offer insights into several prominent practices in LLM research, as summarized in the table below, which has now been added to the manuscript. These findings contribute to the broader goal of explainability by elucidating why certain techniques are effective and motivating the use of coupling as a guiding metric for further exploration and enhancement.
>
> ### Comparison Table: Current Research Practices and Our Contributions
> | **Current Research Practice**                              | **Key Idea**                                                                                     | **Our Contribution**                                                                                   |
> |------------------------------------------------------------|--------------------------------------------------------------------------------------------------|---------------------------------------------------------------------------------------------------------|
> | **Compressing models by merging blocks** [1, 3]          | Combine adjacent transformer blocks to reduce model size without significant performance loss.  | Demonstrates that merging is effective because blocks become strongly coupled during training.          |
> | **Compressing models by pruning blocks** [2,3,4,5]          | Remove certain transformer blocks while preserving functionality.                               | Explains that pruning works because the coupling ensures similarity across blocks.                      |
> | **Compressing models by projecting weight matrices** [6]| Reduce dimensionality by projecting weights into smaller subspaces.                             | Shows that coupling induces a low-dimensional subspace in which blocks' weights are aligned.            |
> | **Studying the effect of transformer block permutations** [7, 8, 9, 12]| Investigate whether permuting the order of blocks affects model performance.                    | Explains why permutations have minimal impact: strong coupling creates structural robustness.           |
> | **Early exiting in LLMs** [10, 11]                         | Allow models to exit computation early based on task confidence.                                | Reveals that early exiting works because representations progress linearly along a shared trajectory due to coupling. |
>
> ___
>
> **“relying solely on potential future utility is not a strong argument in itself. Without practical applications, the concerns about its utility remain inevitably”**
>
> Many phenomena, such as the information bottleneck [13], grokking [14], neural collapse [15], and deep double descent [16], initially emerged as empirical observations without immediate practical applications. Over time, these insights have inspired valuable practical advancements. Similarly, we hope our work on transformer block coupling phenomena will pave the way for future research that enhances transformers, which is enabled by a thorough understanding of their representations.
> ___
>
> [1] Fu et. al. “DepthShrinker: A New Compression Paradigm Towards Boosting Real-Hardware Efficiency of Compact Neural Networks”. 2022
>
> [2] Elkerdawy et. al. “To Filter Prune, or to Layer Prune, That Is The Question”. 2020.
>
> [3] Kim et. al. “LayerMerge: Neural Network Depth Compression through Layer Pruning and Merging”. 2024.
>
> [4] Dror. et. al. “Layer Folding: Neural Network Depth Reduction using Activation Linearization”. 2022.
>
> [5] Fang et. al. “Structural pruning for diffusion models”. 2023.
>
> [6] Ashkboos et. al. “SliceGPT: Compress Large Language Models by Deleting Rows and Columns”. 2024.
>
> [7] Hu et. al. “Lora: Low-rank adaptation of large language models”. 2021.
> [8] Mahabadi et. al. “Compacter: Efficient low-rank hypercomplex adapter layers”. 2021.
>
> [9] Ouderaa et. al. “The llm surgeon”. 2023.
>
> [10] Scardapane et. al. “Why should we add early exits to neural networks?”. 2020.
>
> [11] Jazbec et. al. “Early-Exit Neural Networks with Nested Prediction Sets”. 2024
>
> [12] Li et. al. “Recovery guarantee of weighted low-rank approximation via alternating minimization”. 2016.
>
> [13] Tishby et. al. “Deep learning and the information bottleneck principle.” 2015.
>
> [14] Power et al. “Grokking: Generalization Beyond Overfitting on Small Algorithmic Datasets.” 2022.
>
> [15] Papyan et al. “Prevalence of neural collapse during the terminal phase of deep learning training.” 2020.
>
> [16] Nakkiran et al. “Deep double descent: where bigger models and more data hurt.” 2019.

---

> ### Author Response · Authors · 2024-11-27
>
> **“As the new Figure 39 shows, the correlation between coupling and performance diminishes significantly among modern high-performing LLMs”, “reporting results that do not always strengthen their main claims (like Figure 39).”**
>
> We appreciate the acknowledgment of our scientific transparency.
>
> To ensure a comprehensive analysis, we included models spanning a wide performance range. Restricting the analysis to a subset of high-performing models naturally impacts the observed correlation. Indeed, when focusing on models with benchmark scores above 45, we observe a decrease in the strength of the linear correlation. However, the correlation remains statistically significant, particularly in Figure 39a, as reported in our general comment.
> ___
>
> **“the possible existence of numerous complex metrics that monotonically increase or decrease during training leaves room for skepticism”, “It is difficult to argue that coupling is uniquely explanatory”, “performance of a single downstream task like GSM8K”**
>
> Thank you for raising this important point. We would like to clarify that we do not claim coupling to be uniquely explanatory of generalization. Instead, we argue that it is a significant factor based on our analysis.
>
> To address the specific example about GSM8K, Figure 31 in the manuscript illustrates the correlation between the GSM8K score and the overall LLM benchmark score. While GSM8K performance shows some correlation as one would expect ($R^2 = 0.57$), the correlation between coupling and the LLM benchmark score is notably stronger ($R^2 = 0.75$). This suggests that coupling captures a broader or more intrinsic property of the model’s training dynamics compared to individual downstream tasks or other hyperparameters examined in our study.
> ___
>
> **“It is difficult to argue that coupling [occurs]… across diverse Transformer or LLM behaviors.”**
>
> Our empirical analysis demonstrates coupling's consistent presence through strong statistical correlations between depth-wise coupling strength and generalization. Specifically, we observe an $R^2$ value of $0.75$ with a highly significant p-value of $1.56\times 10^{-6}$ for the full model set. Even when examining specific subsets - such as 7B models ($R^2=0.55$, $p=0.023$) or high-performing models with scores above 45 ($R^2=0.39$, $p=0.023$) - the correlations remain statistically significant at the standard $p<0.05$ threshold. These statistical relationships across different model populations support coupling's consistent emergence across a broad set of transformer and LLM behaviors.
>
> ___
> **“This suggests that coupling may be less robust as a comparative metric across different models or training setups. Instead, it risks being a proxy for secondary factors (e.g., training compute or other unknown factors) rather than capturing an essential underlying mechanism.”**
>
> We appreciate the reviewer’s comment. If coupling is a proxy for a secondary factor which correlates with generalization, then:
> 1. The secondary factor should correlate with coupling; and
> 2. The secondary factor should correlate with generalization better than coupling correlates with generalization.
>
> To explore this possibility, we conducted an extensive series of experiments, analyzing correlations across a wide range of variables:
> - **Generalization correlations:**
>   - Number of parameters (Figure 35a)
>   - Embedding dimension (Figure 35a)
>   - Model depth (Figure 35a)
>   - Number of training tokens (Figure 8b)
>   - GSM8K score (Figure 31)
> - **Coupling correlations:**
>   - Number of parameters (Figure 35b)
>   - Embedding dimension (Figure 35b)
>   - Model depth (Figure 35b)
>   - Training tokens (Figure 8a)
>   - Prompt difficulty (Figure 9a)
>   - Prompt length (Figures 9b, 38b)
>   - Positional encoding methods (Figure 38a)
>
> Across these analyses, coupling consistently shows the strongest correlation with generalization, significantly exceeding all other tested factors. This comprehensive set of experiments supports our conclusion that coupling cannot be easily dismissed through confounding another hyperparameter.
> ___
>
> **“the caption mistakenly refers to "Mistral 30B.”**
>
> Our apologies. We misinterpreted your previous comment. We have updated the caption to indicate that MPT is the correct model and thank the reviewer for this observation.
> ___
>
> We would like to again thank the reviewer for their continued dedication to helping us improve our work. If the reviewer feels inclined, we would greatly appreciate further reconsideration of our score.

---

> > ### Author Response · Authors · 2024-11-30
> >
> > We thank the reviewer for their engagement during the rebuttal process.  We wanted to ensure that the reviewer’s concerns have been addressed, especially given that their initial review was more critical than others. We would greatly appreciate if the reviewer could provide us with some feedback so that we could take further action should the reviewer have any more concerns prior to the deadline. We want to thank the reviewer again for their in-depth review and reiterate that we do very much appreciate the comments that were made in the original review.

---

> ### Author Response · Authors · 2024-12-02
>
> We again thank the reviewer for the thoughtful engagement during the rebuttal process. As the rebuttal phase is nearing its conclusion, we wanted to follow up to ensure that all of the reviewer’s concerns have been fully addressed. While we are no longer able to make edits to the manuscript itself, we remain committed to addressing any remaining issues or clarifying any points to the best of our ability before the deadline.
>
> Once again, we sincerely appreciate the time and effort the reviewer has dedicated to providing detailed comments, and we value the constructive insights.

---

### Official Review · Reviewer_UDjb · 2024-10-26

**Soundness:** 3
**Presentation:** 4
**Contribution:** 2
**Rating:** 6
**Confidence:** 2

**Summary:**

This paper studies the inference dynamics in LLMs, by looking at several properties of the "trajectories" that tokens representations form as they evolve through the network. It is found that the linearized approximations of different blocks in a trained transformer tend to align with each other (in the sense of their top singular vectors being aligned), and that the magnitude of this alignment correlates with overall model performance. Several other properties of the trajectories are also discussed.

**Strengths:**

The paper offers an intriguing observation, expanding on previous works that have reported similar phenomena in resnets. It makes a strong use of the large number of openly available LLMs to study hidden representations. The observation that the "coupling" performance correlates with model performance is of potential interest to the community and could encourage further research.

**Weaknesses:**

The paper seems to be somewhat missing a more coherent organization and message. In particular:
- What is the motivation to consider the additional metrics of "Linearity" (Section 3.2) and "Layer-wise exponential growth" (Sec 3.3)? Can these be related, theoretically or empirically, to the "coupling" phenomena that is the main focus of the paper? (There's a small attempt to answer that in Section 6.2, I don't think it is very satisfying -- see below).
- The coupling phenomena is presented as an "interesting observation", but the paper doesn't offer much more than that, and so the reader is left somewhat unsatisfied. I think it would significantly improve the manuscript if the authors would study this phenomena in a more "functional" way -- is this a uniform behavior for all types of tasks/prompts, or are there some variations that we can understand? Is coupling "important" to either learning or inference, or is it an epiphenomena?
- Another possible way of making the findings more meaningful is a more mechanistic understanding of the phenomena (for example, does it happen in small models, trained on synthetic data? and so on; although this might be further away from the paper's overall approach).

Other than that, there are some "local" issues:

Line 35: The explanation for what a "discrete non-linear dynamical system" is, is pretty confusing. Almost every term seems to be at least inaccurate:
- "The term discrete reflects the network finite depth" (line 38): the term discrete is typically referring to the fact that *time* is discrete in these models, and that the dynamics are described in terms of finite differences rather than rate of change (as would be in a differential equation). It would still be "discrete" even if the network depth would be infinite (i.e., in discrete-time RNNs). The converse is also not true, a continuous-time dynamical system can be evaluated/evolved for finite time.
- "coupled refers to the interdependent token trajectories enabled by self-attention": this is inaccurate, the dynamics will be coupled even in a simple MLP (in fact even in a linear network -- consider the dynamics $\mathbf{x}^{(t+1)}=\mathbf{A}\mathbf{x}^{(t)}$ : for a non-diagonal $\mathbf{A}$, the dynamics is coupled because there are interactions between $x_i$ and $x_j$ through $\mathbf{A}$.)
- "dynamical is due to the residual connections spanning various layers": this is also an odd statement. the term "dynamics" is typically understood simply to refer to the fact that the state is changing with time. Consider the dynamical system from the previous bullet (as an example for dynamical system without the residual connections).
If the authors find it crucial to use non-standard interpretations of standard terms, they should at least note that, and hopefully provide a reasoning for their choice.



Figure 2 and section 5.3: The fact that the rise in "coupling" is so fast seems like a strong indication that the correlation between coupling and performance (as shown in Fig 1) is due to some other variables/confounders, and that coupling in itself has no "causal" importance. The reason is that, presumably, performance decreases gradually and doesn't plateau after 10-20K training steps, like coupling. It could still be the case that coupling early-on facilitates further training (even if not a necessary component of inference in a trained model). In that sense, the authors statement  "this finding suggests that the development of training methods to amplify coupling across transformer blocks may lead to favourable model performance" (Line 431) seems to be a strong claim, as the authors haven't demonstrated any evidence for a causal relationship between coupling and performance.


Section 6.2:
I find the entire explanation rather confusing.
The linearization of $F^l$ in and by itself doesn't mean one can approximate the entire dynamical system as a linear dynamical system. The system is linearized *at* the particular input $x$, so we can't read out much about the dynamics of the same input -- one has to introduce a perturbation in $x$ and *then* use the linearized dynamics to study the evolution of this perturbation (and while it's not clear how to perturb input tokens, it is easy to perturb their embeddings in the first layer).
I would also suggest to the authors to re-write their A.4 appendix in terms of a discrete dynamics (the same arguments can be made, but will be more relatable to the actual content/method of the paper)

**Questions:**

Figure 3: The LSS is defined as $\frac{L}{\lVert \tilde{x}^l_{i}-\tilde{x}^0_{i} \rVert}$ -- clearly a ratio of two positive numbers. How can it be that LSS values reported in Figure 3 are negative?. The authors themselves mention that "Note that $\text{LSS}\geq1$" (in Line 216) which is inconsistent with Fig. 3

Figures 4 and 5: Readability will be greatly improved if some indications of the relevant "axes" is added to the figure itself (Layer/Time in Fig 4/5; embedding dimension on one of the panels; ). Since the different models basically exhibit "identical" results (at least qualitatively), I will advise the authors to keep just one column (one model) for each Fig in the main text, and use the freed space to better visual explanation (other models can be moved to Supplementary).
Even more importantly: a scale/colorbar should be included. Currently it's not even clear if different sub-panels have the same scale or no. The choice to present absolute values is also a bit odd and not well-motivated.

---

> ### Author Response · Authors · 2024-11-22
>
> We thank the reviewer for the thorough feedback, which raised many important questions and thoughtful suggestions. The updated manuscript now features several additions as highlighted in our general message above. Figures (1-3) are clarified to be interpretable, Section 6 has major revisions in response to reviewer comments. Additionally, we have trained 64 ViT with varying stochastic depth and weight decay settings to further analyze the emergence of coupling in transformers (Figures (6, 10, 11)). We also examine how coupling varies across layers, complemented with a comparison of prompts with and without Zero-Shot CoT, as displayed in Figure (15).
>
> The main goal of the paper was to study the trajectories of internal representations in transformer-based LLMs, and explore their relationships with each other. Key properties of these representations include the coupling of transformer blocks, linearity of representations, and exponential spacing. Our contribution establishes a fundamentally new phenomenon in LLMs which consistently occurs across many models which were designed and trained independently by at least 6 distinct organizations. The coupling behavior and regularity in internal representations occur uniformly across models: we are not aware of such a measurement that consistently emerges within all measured open LLMs.
>
> **"What is the motivation to consider the additional metrics of ‘Linearity’ (Section 3.2) and ‘Layer-wise exponential growth’ (Sec 3.3)?” “Can these be related, theoretically or empirically, to the "coupling" phenomena that is the main focus of the paper?“, “Section 6.2: I find the entire explanation rather confusing.”**
>
> We thank the reviewer for the comment and the opportunity to clarify. We have extended the discussion in Section 6.2 (now Section 6.1) with the following to explicitly relate the observed empirical trends of exponential growth and linearity to the theoretical perspective outlined below.
> Suppose that the representations follow the linearized equation:
> $$ x^{l+1} = x^l + J_l x^l = (I + J_l)x^l $$
> Expanding across layers, the entire system can be approximated by the product
> $$ x^L = (I + J_L)(I + J_{L-1})\cdots (I + J_1)x^0  $$
> Assuming that $J_l \approx USU^T$, this equation predicts that the norm of $ x_l $ would exhibit exponential growth layer by layer. Expanding $U$ and $S$,
> $$
>    x^l = \sum_{j=1}^{d_\text{model}} u_j (1+s_j)^l u_j^\top x^0
> $$
> where $ u_j $ and $ s_j$ represent the eigenvectors and eigenvalues of the Jacobian, respectively. The above equation predicts that the norm of $x^l $ would exhibit exponential growth layer by layer. In general, trajectories are not expected to be perfectly linear unless $ x^0 $ aligns with an eigenvector of $ J $. However, in our experiments, we observe a notable tendency towards linearity, suggesting that the representations align progressively during training with the eigenvectors of the coupled Jacobians.

---

> ### Author Response · Authors · 2024-11-22
>
> **“Is this a uniform behavior for all types of tasks/prompts, or are there some variations that we can understand? ”**
>
> The reviewer raises an excellent question, and to investigate it, we have run a series of experiments to compare the coupling measurements across:
> 1. **Task difficulty.** Utilizing the [data] dataset, we have plotted the “difficulty” score against our coupling metric which is displayed in Figure (9a) in the updated manuscript. Based on our measurement of over 500 prompts, no correlation is observed. This may not be surprising, however, since the “difficulty” score of [1] is “human-rated”, and resultantly does not necessarily accurately reflect the difficulty from the perspective of a neural network. In simple terms, what an LLM finds difficult may be different from what a human finds difficult.
> 2. **Token length.** Figure (9b) illustrates the relationship between prompt length and coupling score. The results indicate that coupling tends to decrease as the prompt length increases. This trend implies achieving and maintaining coupling becomes increasingly challenging with a greater number of tokens, likely due to the added complexity of longer contexts.
> 3. **Datasets.** Table (2) in the revised document displays the strength of the correlation between mean coupling task-specific model performance on various tasks and datasets. We hypothesized that coupling strength could increase with model performance, and indeed an arguably weak positive trend can be observed for the models considered.
> 4. **Chain of Thought Ablation.** In Figure (15), we analyze how coupling varies across layers and examine its changes on prompts with and without Zero-Shot Chain of Thought (CoT) [2]. Our results show that the coupling behavior is consistent within model families, with similar layer-wise trends observed across different models of the same family. The CoT prompt largely preserves these trends but introduces slight variations in coupling strength—for example, CoT prompts result in higher coupling in LLaMA models.
>
> The results are now added to the updated manuscript. We believe that these experiments increased the robustness, validity, and relevance of our work, and so we are very thankful to the reviewer for providing this suggestion.
>
> [1] Ding et. al. Easy2Hard-Bench: Standardized Difficulty Labels for Profiling LLM Performance and Generalization. 2024. https://arxiv.org/abs/2409.18433
>
> [2] Kojima et. al.  Large Language Models are Zero-Shot Reasoners. 2023
>
> **“The explanation for what a "discrete non-linear dynamical system" is, is pretty confusing. Almost every term seems to be at least inaccurate”**
>
> This is a motivating statement in our report and sets up the intuition for the entire paper. We truly appreciate the reviewer’s care when considering this statement. We have updated the statement as follows:
>
> >*“Viewing the skip connections as enabling a discrete time step, we represent the hidden representations as dynamically evolving through the layers of the network. The term nonlinear refers to the nonlinear transformations introduced by activation functions, and coupled refers to the interdependent token trajectories that interact through the MLP and self-attention blocks.”*
>
> This is now included in the updated manuscript.

---

> ### Author Response · Authors · 2024-11-22
>
> **“Is coupling "important" to either learning or inference, or is it an epiphenomena?”**
>
> We thank the reviewer for the thought-provoking and interesting question. In response, we have added a subsection with the following discussion to Sections 6.1, 6.2:
>
> *Inference*
>
> > The coupling phenomenon provides insight into the internal operations of transformers. We hypothesize that during training, the LLM learns to represent embeddings in specific low-dimensional subspaces [3, 6]. Given an input, the first layer converts the input into embeddings within one of these learned subspaces. Each subsequent transformer layer modifies these embeddings, potentially moving them to different subspaces. Strong coupling between consecutive layers suggests that the LLM tends towards representations in the same or similar subspaces across many layers. Weak coupling suggests that the subspaces may change between layers, though usually gradually, and that adjacent layers still operate in relatively similar subspaces. Previous works [4] have shown that the early and late layers of language models behave differently, which may be understood through coupling; tokens remain in similar spans, then transition to a different subspace, continuing within a new span that is consistent in the remainder of the transformer.
>
> *Training*
>
> > The emergence of coupling with training steps (Figure 2) may provide insight into the dynamics. Under full coupling and a difference equation approximation, the representations evolve as
> $$
>    x^l = \sum_{j=1}^{d_\text{model}} u_j (1+s_j)^l u_j^\top x^0
> $$
> where $ u_i $ and $ \lambda_i $ denote the eigenvectors and eigenvalues of the Jacobian, respectively. In this case, the gradients of the loss $L$ with respect to prediction $y$, $x^L$ are represented by
> $$ \frac{\partial \text{L}}{\partial x^0} (x^L, y) \approx \sum_{j=1}^{d_\text{model}} u_j (1+s_j)^{L} u_j^\top  (y - x^L) $$
> Due to the coupling, the dynamics exhibit either exponential growth or decay in different subspaces, depending on the sign of $s_j$, which is known to cause challenges for optimization as in past works on dynamical isometry [5]. From the above, we infer that increasing coupling during training makes optimization progressively more difficult. Conversely, as training progresses, it becomes harder to achieve stronger coupling, and is consistent with the logarithmic trend in Figure (1).
>
> A new PCA animation, available in the following anonymized link [7], provides visual evidence supporting this hypothesis: tokens initially move within a specific low-dimensional subspace for several layers. At a certain point, they transition to a different subspace, continuing their trajectory within this new subspace for the remainder of the model.
>
> [3] Eldar et. al.Robust Recovery of Signals From a Structured Union of Subspaces. 2009
>
> [4] Lad et. al. The Remarkable Robustness of LLMs: Stages of Inference? 2024.
>
> [5] Pennington et. al. Resurrecting the sigmoid in deep learning through dynamical isometry: theory and practice. 2017
>
> [6] https://dhpark22.github.io/images/unionsubspaces.png
>
> [7] https://drive.google.com/file/d/1bcvQs6dKIOzyutmwXdLA0NGQ8_7tssE_/view?usp=sharing
>
> **“Another possible way of making the findings more meaningful is a more mechanistic understanding of the phenomena (for example, does it happen in small models, trained on synthetic data?”**
>
> We thank the reviewer for suggesting these additional experiments. Figure 4 now demonstrates the presence of coupling in 64 small ViT classifiers on CIFAR10, with varied weight decay and stochastic depth. We observe that coupling correlates with test accuracy when each stochastic depth value is fixed (Figure 4a). In addition, stochastic depth improves coupling (Figure 4b) for fixed weight decay values. We hypothesize that stochastic depth transformer blocks to be operationally similar and consequently improves coupling.
>
> Further discussion of ViT training is included in Appendix A6 and Figures (10, 11).
>
> **“Figure 2 and section 5.3: The fact that the rise in "coupling" is so fast seems like a strong indication that the correlation between coupling and performance (as shown in Fig 1) is due to some other variables/confounders.”**
>
> Figure (1b) in the updated document displays an updated plot of the normalized coupling as a function of training which is now plotted with logarithmically-spaced checkpoints. This figure displays a  monotonic and consistent growth in coupling strength throughout training. Although the rate at which coupling improves decreases as a function of training checkpoints, this may be expected as discussed above.

---

> ### Author Response · Authors · 2024-11-22
>
> **“The authors statement "this finding suggests that the development of training methods to amplify coupling across transformer blocks may lead to favorable model performance" (Line 431) seems to be a strong claim, as the authors haven't demonstrated any evidence for a causal relationship between coupling and performance.”**
>
> We thank the reviewer for this remark. In response, Line 461 now reads
>
> > developing training methods to amplify coupling across transformer blocks could provide additional regularization and improve performance
>
> While we agree that we cannot conclude a causal relationship from our measurements, our results display a statistically significant positive correlation.
>
> **“Section 6.2: I find the entire explanation rather confusing. The linearization of $F^l$ in and by itself doesn't mean one can approximate the entire dynamical system as a linear dynamical system.”**
>
> We thank the reviewer for the opportunity to clarify our discussion. Indeed, we are not asserting that the entire dynamical system is a linear dynamical system. Rather, we argue that when the dynamical system is restricted to a local neighborhood in the input space, the Jacobians across the depth of all layers exhibit coupling. As a result, at a specific point within this neighborhood, the forward pass of the model can be effectively approximated as a linear dynamical system. We have updated the text to reflect these nuances and provide additional context to make this distinction more explicit.
>
> **“The system is linearized at the particular input $x$, so we can't read out much about the dynamics of the same input -- one has to introduce a perturbation in $x$ and then use the linearized dynamics to study the evolution of this perturbation (and while it's not clear how to perturb input tokens, it is easy to perturb their embeddings in the first layer).”**
>
> We appreciate the reviewer's insightful comment. Indeed, the system is linearized at a particular input $x$, and at this point, the Jacobians of various layers exhibit coupling. This coupling provides meaningful insights into the dynamics associated with the specific input $x$, even in the absence of explicit perturbations.
>
> The reviewer raises an excellent point: analyzing perturbations around this input is an essential approach to gain deeper insights. To address this, we have analyzed the stability of Llama-3 8B by perturbing the embeddings. Figure (32) in Appendix A7 demonstrates the stability of an LLM to perturbations. Perturbations are especially valuable as they relate directly to the model's ability to generalize. The stability of the system under small (or large) perturbations in the vicinity of $x$ offers important clues about its robustness and generalization capabilities.
>
> **“I would also suggest to the authors to re-write their A.4 appendix in terms of a discrete dynamics”**
>
> We thank the author for the recommendation. Section 6.2 and Appendix A.4 are now revised in terms of discrete dynamics, and feature improved justification for the selection of our metrics.
>
> **“How can it be that LSS values reported in Figure 3 are negative?”**
>
> We thank the reviewer for suggesting this improvement to the new Figure 3. The LSS is a positive scalar as defined in Section 3.2. In Figure 3 (now figure 28),  however, we are plotting the linearity of the trajectories, which we define to be (-1) * LSS. This choice was made since increasing LSS corresponds to decreasing linearity, which was a possible point of confusion in interpreting the plot. We have further clarified in the Figure caption.
>
> **“Figures 4 and 5: Readability will be greatly improved if some indications of the relevant "axes" is added to the figure itself”**
>
> Thank you for this recommendation, this vastly improves the interpretation of these figures. We have updated Figures (3 and 4) in the manuscript to include both axes and a color bar. Since distinct Jacobians have differing singular values, the upper bound between each subplot varies, but there is a common lower bound of 0. Additional plots are now included in Appendix A7.
>
> **The choice to present absolute values is also a bit odd and not well-motivated.**
> We chose to present absolute values in the plotted matrices because our primary focus is on how close the matrix is to a diagonal matrix. Using absolute values allows for a clearer visual representation, where entries close to zero can be easily identified, with the minimum color value set to zero rather than a negative value. This choice helps emphasize the degree of diagonalization more effectively.
>
> We again reiterate our appreciation to the reviewer for their detailed recommendations which led to revisions that improved the quality and clarity of our paper. In light of these refinements, should the reviewer find it fitting, we would be most grateful for any potential reconsideration of our score.

---

> ### Comment · Reviewer_UDjb · 2024-11-25
>
> The author response has improved some aspects of the manuscript (in particular, new Fig. 2, the improvement to Figs 3&4, the revised description of the dynamical systems background). To reflect this I have updated the "presentation" score to 4.
>
> The conceptual limitations largely remain open. The authors provide some hypothesizing for the "functional" implication of the coupling phenomena in training and inference, but this remains speculative at current stage. As for the speed at which the coupling emerges, the point was not that coupling is completely flat but that the dynamics are very fast, so in order to provide a new angle for this the thing that should at least be compared is the evolution of the coupling vs evolution of some "performance" measure. Simply plotting the evolution of the coupling on a logarithmic time-axis is not much more informative than it was before (this is not to say that, by and for itself, the plotting of this on logarithmic time axis is "wrong").
>
> The discussion of the exponential norm growth is more clear now but I think it is still misleading. You can expect exponential growth (or decay) of the norm even without any coupling/alignment "beyond chance" (as a quick example this can be easily simulated by taking $x_0 \sim \frac{1}{\sqrt{d}}\mathcal{N}(0,I)$, but **not** "properly" scaling the matrices, i.e., $W^t_{ij} \sim \mathcal{N}(0,1)$ and setting $x_t=W^tx_{t-1}$.)

---

> ### Author Response · Authors · 2024-11-28
>
> We thank the reviewer for their additional feedback and engagement during the rebuttal process.
> ___
> **“The authors provide some hypothesizing for the "functional" implication of the coupling phenomena in training and inference, but this remains speculative at current stage”**
>
> We acknowledge that the discussion on inference implications is of speculative nature. To address this, we have moved it to the Appendix and toned down the language to highlight the conjunctive nature of the claims.
>
> However, the discussion on training is grounded in objective results, derived under the assumption of coupling.
> ___
> **“in order to provide a new angle for this the thing that should at least be compared is the evolution of the coupling vs evolution of some "performance" measure.”**
>
> We appreciate the reviewer’s valuable feedback. In our Figure 1b, we observe that coupling increases gradually at logarithmically spaced checkpoints. This aligns with the gradual growth of accuracy observed in Pythia 12B and Pythia 6.9B, which also follows a logarithmic trend during training (see this [1] side-by-side comparison).
>
> To address the reviewer’s comment, we have added the following discussion to Section 5.3 of the manuscript:
>
> > The gradual growth of coupling observed in Figure 1b parallels the logarithmic increase in accuracy during training for Pythia 12B and Pythia 6.9B [2]. This highlights the relationship between coupling and performance, since both properties emerge at similar training iterations and rates.
>
> Additionally, we have updated the manuscript to emphasize the following key observations:
>
> 1. **Coupling Growth:** Under **logarithmically spaced checkpoints**, the growth of coupling is gradual rather than immediate, as seen in Figure 1b.
>
> 2. **Performance Increase:** Performance improvement occurs gradually under **logarithmically spaced checkpoints**, consistent with previously observed trends [2].
>
> These insights further illustrate that increasing coupling becomes exponentially harder as training progresses, underscoring its connection to the learning dynamics of large language models.
>
> [1] https://drive.google.com/file/d/10ZDWO6tOjmZxU_JRhkeagoOLhu_IRf53/view?usp=sharing
> [2] Biderman et. al. “Pythia: A Suite for Analyzing Large Language Models Across Training and Scaling”. 2023.
>
> ___
> **“You can expect exponential growth (or decay) of the norm even without any coupling/alignment ‘beyond chance’”
> “This can be easily simulated by taking $x_0 \sim \mathcal{N}(0, I) / sqrt(d)$, but not "properly" scaling the matrices.”**
>
> We thank the reviewer for this comment. While it is certainly true that exponential growth can occur without coupling (e.g., in the case of improperly scaled matrices), such examples do not correspond to a well-trained model. In our work we do not claim that exponential growth arises exclusively from coupling. As detailed in Section 6.1, we show that under complete coupling and regularity of singular values across depth, exponential growth in representations arises and aligns with empirical observations of trained models. This distinction reinforces the relevance of coupling in understanding this phenomenon.
> ___
> We would like to again thank the reviewer for their continued dedication to helping us improve our work. If the reviewer feels inclined, we would greatly appreciate further reconsideration of our score.

---

> > ### Author Response · Authors · 2024-11-30
> >
> > We thank the reviewer for their engagement during the rebuttal process. We would greatly appreciate it if the reviewer could provide us with additional feedback so that we could take further action should the reviewer have any more concerns prior to the deadline. We want to thank the reviewer again for their detailed suggestions and reiterate that we greatly appreciate the comments that were made in the original review.

---

> ### Author Response · Authors · 2024-12-02
>
> We would like to again thank the reviewer for their past engagement during the rebuttal process. We wanted to ensure that the reviewer’s comments have been fully addressed since the discussion period is ending today.  One result that we would like to highlight is the following figure showing the gradual growth of accuracy observed in Pythia 12B and Pythia 6.9B, which follows the same logarithmic trend as coupling during training:
> https://drive.google.com/file/d/10ZDWO6tOjmZxU_JRhkeagoOLhu_IRf53/view?usp=sharing
>
> If the reviewer has any additional questions or concerns before the end of the review process, we would be more than happy to continue the discussion.

---

### Official Review · Reviewer_n94A · 2024-11-03

**Soundness:** 3
**Presentation:** 3
**Contribution:** 3
**Rating:** 8
**Confidence:** 4

**Summary:**

the paper analyzes and correlates jacobian singular value decompositions cross tokens and cross layer-depths. as a result, they define a 'coupling metric', and show that high coupling apparently correlates to high benchmark values.
as such some intel on the inner processings of llm's is generated but the interpretation is widely left to the user.

**Strengths:**

- the paper offers a new perspective on by coupling training dynamics with model performance.
- the method demonstrates somewhat robust as shown with the regression fit
- potentially high impact on model architecture/ training and diagnosis (if better understood)

**Weaknesses:**

the interpretation of this metric is vastly unclear. it is kinda clear that model training dynamics converge over training steps, in particular with common decaying lr-schedulers applied. so i wonder rather if one can interpret the score as a 'model training convergence rate' rather then performance-metric. falcon-7b (the entire model cluster > -.7 in fig1) demonstrates that high-coupling does not necessarily correlate with openllm-benchmarks afterall, tracing the activations/ gradients already shows such a convergence as well. afterall, on a converged model, activations only nuanced shift between layers.

similarly i'm unsure how to interpret fig 6 etc.
computational costs of the methods are also not discussed (?).

overall i appreciate the solid work and do think there is merit in this methodology per se. however more focus should be made to practical implications of this rather abstract metric, and rigorously discussed.

**Questions:**

1) what is the value of fig 6/9?
2) what are your computational costs?
3) anything surprisingly found in the jacobians?

---

> ### Author Response · Authors · 2024-11-22
>
> We thank the reviewer for their comments on our manuscript. To clarify, our work studies the evolution of representations of the LLM from input to output. Properties of these representations include the coupling of transformer blocks, the linearity of representations as a function of depth, and the layer-wise exponential spacing. Coupling is measured on pre-trained models, by computing the Jacobians of the transformer blocks with respect to the inputs of each block, and comparing their singular vectors. Coupling measures the extent to which the Jacobians operate on the same basis.
>
> We analyze this transformer block coupling phenomenon in many openly available LLMs trained by 7 distinct organizations, and we additionally demonstrate its presence through newly added ablation studies on 64 ViTs.
>
> **“The interpretation of this metric is vastly unclear.”**
>
> We thank the reviewer for the feedback on this aspect of our work. To amend this, we have:
>
> - Updated Figure 2. This figure contains a schematic diagram that visually displays the interpretable relationships that are captured by the coupling metric.
> - Revised Section 3.1. The description of the coupling metric is now updated to provide further detail about the measurement.
> - Updated Figure 1.  Figures 1(c-e) now present a new visualization which displays the strength of the coupling between pairs of LLM layers. The new diagram improves intuition when contemplating the purpose of our measurements.
>
> These updates are viewable in the attached updated manuscript. We hope that this will increase the interpretability of our measurements, provide additional clarity to the reader, and highlight the significance of our results.
>
> **“I wonder if one can interpret the score as a 'model training convergence rate' rather than performance-metric. ”**
>
> Indeed, the coupling score increases with training, and is therefore indicative of ‘model training convergence.’ However, it is not limited to being a convergence indicator. The coupling score is strongly correlated with performance, as demonstrated in Figure (1), where higher coupling consistently aligns with improved model accuracy and generalization. This dual role suggests that the coupling score provides a holistic view of the model's behavior, encapsulating both its optimization dynamics and eventual performance.
> Figure (1b) now further clarifies the relationship between the coupling score and `model training convergence’. Specifically, we computed the coupling metric across a broader range of prompts and analyses at logarithmically spaced checkpoints.
>
> **“I'm unsure how to interpret fig 6 etc”, “on a converged model, activations only nuanced shift between layers”, “what is the value of fig 6/9?”**
>
> Figures (12, 13 and 14) (originally 12 and 14 were 6 and 9) illustrate the trajectories of hidden representations as they evolve through the model, revealing a consistent and structured pattern in LLMs that has been overlooked in prior works. These figures visually demonstrate the nuanced shifts in activations between layers, as noted by the reviewer. Importantly, this nuanced transformation is not random but occurs in a highly aligned direction, since consecutive layers move roughly in the same direction. To provide a more precise analysis, Figure (5) measures the LSS and expodistance metrics to quantify the regularity present at consecutive layers. Figures (12, 13 and 14) are therefore highly complementary of the numerical findings, offering an intuitive visualization of the behaviors that the LSS and expodistance metrics capture in detail.
>
> To further address the reviewer’s feedback, we have included a GIF that visualizes the evolution of activations across the model's depth. This GIF, accessible via the anonymized link [1] provided, offers an additional dynamic perspective to enhance understanding.
> [1] https://drive.google.com/file/d/1bcvQs6dKIOzyutmwXdLA0NGQ8_7tssE_/view?usp=sharing

---

> ### Author Response · Authors · 2024-11-22
>
> **“What are your computational costs?”**
>
> Thank you for highlighting this oversight in our report. We have added the following description of the computational costs for each experiment to Appendix A.5:
>
> Coupling: Coupling requires computing the Jacobians for each transformer block, and so a forward pass and backward pass required (note that we compute the Jacobians on a block level). Once the Jacobians are obtained, it requires computing a truncated singular value decomposition of each Jacobian. The time complexity of computing the truncated SVD of rank $k$ for a $d\times d$ matrix is $\mathcal{O}(d^2k)$, where $k << d$. Computing $A$ from the SVDs then has time complexity $\mathcal{O}(k^3)$, so the asymptotic time complexity of computing the coupling score between two connections is $\mathcal{O}(d^2k)$ (in addition to the Jacobian computation).
>
> LSS: For each trajectory, the time complexity of computing the LSS is $\mathcal{O}(Ld)$ where $L$ is the number of layers and $d$ is the hidden dimension. Therefore, for a prompt containing $T$ tokens, the total time complexity for each prompt is $\mathcal{O}(TLd)$ (in addition to a single forward pass of the model).
>
> Expodistance: Similarly, computing the expodistance of a single trajectory has time complexity $\mathcal{O}(Ld)$. Therefore, for a prompt containing $T$ tokens, the total time complexity for each prompt is $\mathcal{O}(TLd)$ (in addition to a single forward pass of the model).
>
> **“Overall i appreciate the solid work and do think there is merit in this methodology”**
>
> We truly appreciate this feedback. We hope that our updates, clarifications, and comments address the interpretability and highlight the significance of our work. If the reviewer is inclined, we would be greatly appreciative of an increase in score.

---

> ### Comment · Reviewer_n94A · 2024-11-25
>
> thank you for your thorough update.
>
> while the results are indeed pretty interesting, unfortunately i got increasingly confused by the added experiments.
>
> 1) still my main claim:  for language models your coupling correlation does not linearly correlate well to model performance. the only reason it seems to, is your random inclusion of various model sizes in fig 1a. i could not find exact values of coupling in your results tables, but reproducing the R value of fig 1a with only 7b's gives me a R^2 value of 0,3, without pythia of <0,1.
> so still: i do not know what you found, its interesting, but its not unconditionally a language model performance correlation - 1b also shows (slightly) higher coupling scores for the 7b, even though the 12b achieves much higher scores on the converged model. perhaps it is a good indicator/ approximation w.r.t. some tolerance though **this still needs to be adressed.**
>
>
> 2) i'm pretty confused why you added vit's now. it seems like a pretty big change at this stage. on this set of experiments of fig6, i could agree (for sd 0,3) that coupling seemingly correlates with performance, when only sweeping the layer dropout. albeit not knowing why the performance is so low - on cifar-10 one easily gets into the 90% regime afaik, perhaps your metric correlates well in the 'lesser converged' regime
>
> ---
> i do agree that the evolvement of activations through the model has not yet been shown so directly, so slightly improved the score for now

---

> ### Author Response · Authors · 2024-11-27
>
> We thank the reviewer for their additional feedback and engagement during the rebuttal process.
>
> **“your coupling correlation does not linearly correlate well to model performance. the only reason it seems to, is your random inclusion of various model sizes in fig 1a.” “its not unconditionally a language model performance correlation”**
>
> We appreciate the reviewer’s observation regarding the inclusion of various model sizes in Figure 1a. While we acknowledge that focusing on a single model size could remove a potential extraneous variable, our intention in including models of various sizes was to examine the coupling correlation across a broader spectrum of models, thereby providing insights into its generalizability.
>
> To address the concern, we conducted additional analyses to isolate the relationship within subsets of the data. Specifically, we computed the coupling correlation for models of a single size (7B) and for models scoring above 45, as well as across the entire dataset. The results are as follows:
>
> | Subset          | $R^2$ | $p$-value  |
> |------------------|---------|----------|
> | All models       | $0.75$    | $1.56\times 10^{-6}$ |
> | 7B models only   | $0.55$   | $0.023$   |
> | Score > 45       | $0.39$    | $0.023$    |
>
> These results indicate statistically significant correlations in all cases, including within a single model size and performance band. While the correlation strength varies, the consistent significance across subsets supports the validity of the relationship, even when controlling for model size.
>
> This reinforces our claim that coupling correlation provides meaningful insights into model performance trends, independent of size variability.

---

> > ### Author Response · Authors · 2024-11-27
> >
> > **“I could not find exact values of coupling in your results tables, but reproducing the R value of fig 1a with only 7b's gives me a R^2 value of 0,3, without pythia of <0,1.”**
> >
> > We appreciate the reviewer’s suggestion. When focusing solely on 7b models, we obtain an R^2 value of 0.55, which is weaker than the overall trend but still represents a nontrivial relationship. As the reviewer suggests, we can remove Pythia, which reduces the R^2 value further to 0.24. However, following this line of reasoning, one could also consider removing Falcon (which appears to be an outlier) to obtain an R^2 value of 0.75.
> >
> > **“1b also shows (slightly) higher coupling scores for the 7b, even though the 12b achieves much higher scores on the converged model. perhaps it is a good indicator/ approximation w.r.t. some tolerance though this still needs to be addressed.”**
> >
> > Indeed, Figure 1b shows marginally stronger coupling for Pythia 6.9b compared to 12b, which initially puzzled us as well, given the expectation that the 12b model would achieve significantly higher scores.
> >
> > Analyzing coupling throughout training required running several checkpointed models. We therefore limited our measurements for this figure to a single dataset, namely ARC, to focus on the general behavior of coupling during training. Interestingly, on ARC, the 12b model achieves an accuracy of 39.59%, while the 7b model performs slightly better with an accuracy of 40.1%. This result aligns with our results, as by the end of training, the 6.9b model exhibits marginally stronger coupling and slightly higher accuracy on the dataset used for coupling measurement. Nonetheless, the primary purpose of Figure 1b is to illustrate the evolution of coupling during training, rather than to establish a direct correlation between coupling and final model performance (which is instead done in Figure 1a).
> >
> > These variations demonstrate the complexity of the relationship and suggest that coupling may be a meaningful but not sole contributor to performance, aligning with our original claim.
> >
> > | Benchmark | Pythia 12b | Pythia 6.9b |
> > |-------------|------------|-------------|
> > | ARC | 39.59 | **40.1** |
> > | HellaSwag | **68.82** | 65 |
> > | MMLU | **26.76** | 24.64 |
> > | TruthfulQA | 31.85 | **32.85** |
> > | Winogrande | 64.17 | **64.72** |
> > | GSM8K | **1.74** | 1.06 |

---

> > > ### Author Response · Authors · 2024-11-27
> > >
> > > **“unfortunately i got increasingly confused by the added experiments” “i'm pretty confused why you added vit's now. it seems like a pretty big change at this stage.”**
> > >
> > > The primary focus of our work remains on transformer block coupling and the regularity of internal representations in pretrained LLMs. The ViT experiments were introduced as a supplemental analysis to explore the generality of this phenomenon across transformer-based architectures. This addition was motivated by feedback from Reviewer UDjb, who recommended examining whether the observed behavior extends to smaller or alternative models to gain a deeper mechanistic understanding of the phenomenon.
> > >
> > > ViTs were selected during the rebuttal phase due to their relevance as a transformer-based architecture, their compatibility with our computational resources, and their alignment with the scope of the suggested experiments. These experiments allow us to investigate the impact of factors such as stochastic depth and weight decay on coupling behavior, providing further evidence for the broader applicability of our findings.
> > >
> > > To address concerns about the perceived magnitude of this addition, we have moved the ViT results to the Appendix. This ensures that the primary focus of the paper remains on pretrained LLMs while still offering interested readers a broader perspective on the phenomenon.
> > >
> > > **“on cifar-10 one easily gets into the 90% regime”**
> > >
> > > The low accuracy can be attributed to the small size of the ViTs (Appendix A3), which was chosen in response to reviewer UDjb. The models have converged after training (500 epochs), however, thorough refinements to training were not carefully performed due to the large number of experiments and limited rebuttal time.
> > > ___
> > > We would like to again thank the reviewer for their continued dedication to helping us improve our work. If the reviewer feels inclined, we would greatly appreciate further reconsideration of our score.

---

> > > > ### Comment · Reviewer_n94A · 2024-11-28
> > > >
> > > > assuming these discussions find their way into the paper, i further increased my score

---

> > > > > ### Author Response · Authors · 2024-11-30
> > > > >
> > > > > We thank the reviewer for the further reconsideration of our score. The latest comments above have already been added to the manuscript. We greatly appreciate the reviewer's involvement since this feedback significantly improved the quality of our work.
> > > > >
> > > > > If the reviewer have any other questions or concerns before the end of the review process, we would be more than happy to continue the discussion.

---

### Official Review · Reviewer_bxkM · 2024-11-04

**Soundness:** 3
**Presentation:** 3
**Contribution:** 3
**Rating:** 6
**Confidence:** 2

**Summary:**

This paper presents an insightful exploration into the internal dynamics of Large Language Models (LLMs). The authors propose a novel framework to evaluate the coupling of transformer blocks through their Jacobian matrices, offering a structured perspective on their inter-token and cross-layer relationships. The study concludes that this coupling is positively correlated with model generalization performance and appears to be a more significant factor than other hyperparameters such as parameter budget and model depth.

**Strengths:**

- The concept of using Jacobians to study transformer block coupling provides a fresh perspective on understanding the internal mechanics of LLMs.

- The authors conduct an extensive empirical analysis across a variety of LLMs, lending credence to their hypothesis regarding the significance of transformer block coupling.

- The paper is well-organized, with clear definitions and logical flow, making it easy to follow the complex concepts and their implications.

**Weaknesses:**

- The most concern lies in the strength of the generalization claims made by the authors. The paper presents empirical evidence suggesting a strong correlation between transformer block coupling and model performance on Open LLM LeaderBoard. However, it seems that some new models (e.g., LLaMA 3, Phi-2) which involve more pertaining steps (tokens) may have a more significant increase in performance than coupling. Revisiting the Figure 2, the coupling seems to emerge at certain training steps and then remain plain. So if the correlation only holds on a limited scope of pretraining steps?

- The model performance is evaluated on Open LLM LeaderBoard. However, for some reasoning tasks, there may be some emergent capabilities [1] that only larger models have. In such settings, is the correlation between coupling and model performance still holding?

[1] Chain-of-Thought Prompting Elicits Reasoning in Large Language Models

**Questions:**

See weaknesses section above.

---

> ### Author Response · Authors · 2024-11-22
>
> We thank the reviewer for the thorough and insightful feedback that raised some important questions and suggestions. We have addressed the reviewer’s suggestions below, and have included the revisions in the updated manuscript.
>
> **“It seems that some new models (e.g., LLaMA 3, Phi-2) which involve more pertaining steps (tokens) may have a more significant increase in performance than coupling”**
>
> We thank the reviewer for this insightful remark, which allowed us to extend our analysis. In our revised manuscript, we have included a plot (Figure (7)) with an exponential fit to account for this behavior, slightly improving the R^2 value. To address this further, we have added Figure (8a) to the Appendix, illustrating the relationship between training tokens and performance. The addition proposed by the reviewer further underscores the significance of our measurements.
>
> Additionally, Figure (8b) now plots the number of training tokens against the benchmark score, showing no clear trend. The results demonstrate that performance exhibits a stronger correlation with coupling than with the number of training tokens, providing further support for our findings.
>
> Furthermore, we emphasize that the correlation between coupling and benchmark performance is significantly stronger than that of other relevant hyperparameters, as illustrated in Figure (35) in the Appendix.
>
> **“Revisiting the Figure 2, the coupling seems to emerge at certain training steps and then remain plain. So if the correlation only holds on a limited scope of pretraining steps?”**
>
> The reviewer raises an excellent point about the clarity of the trend in Figure 2. Figure 1b is now updated with logarithmic spacing of the training checkpoints and evaluated on additional prompts. The new figure highlights a monotonic and consistent growth in coupling strength with training. This suggests that coupling increases with training steps and that the trend is approximately logarithmic.
>
> **“The model performance is evaluated on Open LLM LeaderBoard. However, for some reasoning tasks, there may be some emergent capabilities [1] that only larger models have. In such settings, is the correlation between coupling and model performance still holding?”**
>
> Indeed, it is interesting to explore how coupling behaves in the presence of techniques such as Chain of Thought (CoT) [1]. To address this, we conducted a more in-depth analysis, as shown in Figure 15 of the updated manuscript, by examining how coupling varies across layers, complemented with a comparison between prompts with and without Zero-Shot CoT [2].
>
> Our findings indicate distinct coupling behaviors across models, but with similar layer-wise trends across different models within model families. In the LLaMA models, coupling starts low, peaks in the middle layers, decreases, and then slightly increases at the final layers. Gemma models exhibit a high initial coupling that steadily decreases toward the end, while Phi models display low coupling in the first layer, an immediate increase, and a gradual decline in later layers. The CoT prompts largely preserve these trends but with slight variations in coupling strength. For LLaMA models, CoT prompts consistently result in higher coupling across layers. In Gemma models, CoT prompts yield similar coupling overall, with minor variations across layers. In the Phi models, Phi-2 shows lower coupling with CoT prompts, while Phi-1.5 exhibits a marginally higher coupling. These variations in behavior, alongside the similarities within model families, likely stem from differences in training methods and data across organizations, while models within the same family may be trained using similar methodologies.
>
> [1] Wei, Jason et al. “Chain of Thought Prompting Elicits Reasoning in Large Language Models.” (2022).
>
> [2] Kojima et. al.  Large Language Models are Zero-Shot Reasoners. 2023
>
> We truly appreciate the reviewer’s thoughtful feedback and hope that our additions, clarifications, and analyses have adequately addressed the concerns raised while also enhancing the significance of our work. If the reviewer finds our responses satisfactory, we would be sincerely grateful for a reconsideration and potential increase in score.

---

> ### Author Response · Authors · 2024-11-26
>
> Dear reviewer,
>
> Apologies for the repeated follow-ups, but as the deadline for editing the manuscript **quickly** approaches, we want to ensure that all your concerns have been addressed. We would greatly appreciate any additional feedback you might have so that we can make any necessary revisions before the deadline.
>
> We are sincerely grateful for your initial comments and look forward to further discussion if there are any remaining concerns.

---

> > ### Comment · Reviewer_bxkM · 2024-11-28
> > **Response to Authors**
> >
> > Dear Authors,
> >
> > Thanks for your response and efforts. My concerns have been mostly addressed, but the analyses about CoT across different models demonstrate various behaviours. This may indicate that transformer block coupling may be one of the implicit features that affect model performance, which is sensitive to training and inference strategies which are diverse in current LLMs (e.g., O1). Generally, I'll keep my positive score, but I do not have enough confidence to raise it.

---

> ### Author Response · Authors · 2024-12-02
>
> We thank the reviewer for their additional feedback and engagement during the rebuttal process.
>
> We initially plotted the coupling for each layer to gain a deeper understanding of how coupling varies across layers, observing that coupling generally decreases from the middle to later layers, with even stronger consistency of depth-wise patterns for models of the same family. To enable more direct comparisons on the effects of CoT on coupling across different models, we compute the average coupling across all layers and incorporate additional prompts, including those that may benefit from Chain of Thought (CoT) prompting and those that may not.
>
> **“the analyses about CoT across different models demonstrate various behaviours”**
>
> We appreciate this observation and have refined our chain-of-thought (CoT) experiments for greater consistency. Our updated results indicate an increase in mean normalized depth-wise coupling with CoT prompting, presented in the following table:
>
> | Model   | With CoT | Without CoT |
> |---------|---------------------------|------------------------------|
> | Qwen 2 0.5B   | **0.0745**             | 0.0675                       |
> | Llama-3 8B  | **0.1072**                | 0.0894                       |
> | Phi 2B    | **0.1593**                      | 0.1439                       |
> | MPT 7B   | **0.1007**                    | 0.0990                       |
> | Llama-2 7B  | **0.1197**                | 0.0872                       |
> | Gemma 2B   | 0.1686                    | **0.1793**                   |
>
> Since CoT techniques generally enhance performance, these findings support the hypothesis that stronger coupling correlates with stronger model performance. Some differences arise partially from the fact that coupling was computed using the same set of prompts across all models, and that there is variability in the models that improve with CoT on these prompts. We do note an exception in the case of Gemma 2B, where coupling slightly decreases with CoT prompting. This deviation may arise from model-specific factors, such as architectural differences, sensitivity to the chosen prompts, or underlying training dynamics. These results have now been added to Appendix A8 of the manuscript.
> ___
>
> **“coupling may be one of the implicit features that affect model performance”, “sensitive to training and inference strategies”**
>
> We agree that model architecture, training, and inference strategies contribute to the observed variability. While our results fairly consistently show that increased coupling aligns with better performance, the magnitude of this effect is model-dependent. This underscores the complexity of coupling as a performance indicator and its sensitivity to various design choices.
> ___
> Although this analysis is not central to our paper, we hope it provides clarity on CoT-related behaviors and supports broader discussions in the literature. Please do not hesitate to share any further questions or feedback before the rebuttal deadline.

---

### Author Response · Authors · 2024-11-24
**Overview of Revisions**

Dear Reviewers and AC,

Thank you for your extensive and insightful feedback. We have uploaded a revised version of the manuscript, and the changes have been discussed in our responses to the reviews. Below, we summarize the key revisions:

___

## **Major Revisions**

1. **New Figure 1.** A plot containing key findings of the coupling phenomenon. We have improved Figure 1b demonstrating the emergence of coupling with training. Additionally, this figure contains a new visualization: an adjacency plot displaying the pair-wise strength of depth-wise coupling between layers at various training checkpoints. This plot demonstrates the layer-wise locality in terms of the strength of coupling.
2. **New Figure 2.** A brand new schematic diagram that illustrates the coupling measurement. We believe that this diagram significantly improves the interpretability and digestibility of our work.
3. **Improved Section 6.1.** We have better motivated the regularity being a result of coupling following the observations of one of our reviewers. Additional discussion for the choice of metrics is included in Appendix A4.
4. **Revised Section 6.2.** We provide a better motivated discussion for the presence of coupling and relation to past works.

___

## **Minor Revisions**

1. **Clarification of perspective.** We have revised the introduction of our dynamical systems perspective to be more precise to our true interpretation.
2. **Revised Section 3.1.** We have revised this section to enhance the interpretability of our experiments by providing a clearer presentation of the motivation and derivation behind our coupling measurements.
3. **Updated Figures 3 & 4.** Although these still contain the same information, labels of the relevant axes and a color bar have been included to increase their readability.
4. **Merged Figures 5 and 6.** We have merged the linearity and exponential distance plots, and have included the measurements for the untrained variants in both.
5. **New Figure 6.** We train 64 small ViTs on CIFAR10 and measure the effect of training hyperparameters (stochastic depth, weight-decay) on coupling and accuracy (see Section 4.3). At fixed stochastic depths, we find that coupling positively correlates with performance (Figure 4a) and coupling improves with greater stochastic depth (Figure 4b).
6. **New Figure 15.** We analyzed how coupling varies across layers and examined its changes on prompts with and without Zero-Shot Chain of Thought (CoT). Our results show that the coupling behavior is consistent within model families, with similar layer-wise trends observed across different models of the same family. The CoT prompt largely preserves these trends but introduces slight variations in coupling strength—for example, CoT prompts result in higher coupling in LLaMA models.
7. **Many additional experiments.** Following the feedback from the reviewers, many additional experiments have been undergone, the results of which have been included in the appendix of the updated manuscript.
___
We would again like to thank the reviewers and the AC for their thoughtful consideration of our work. We welcome any further feedback on our rebuttal.

---

### Author Response · Authors · 2024-11-28
**Regarding Correlation with Generalization**

Dear Reviewers and AC,

We thank the reviewers for their continued dedication to improving our manuscript. Thus far, the review process has raised concern regarding the strength of the linear correlation between the strength of coupling and the HuggingFace LLM Benchmark Score (Figure 1a in the manuscript), particularly when restricting to certain subsets of the data. To further clarify this point, we present the following table summarizing the results across various subsets:
___
| Subset          | $R^2$ | $p$-value  |
|------------------|---------|----------|
| All models        | $0.75$    | $1.56\times 10^{-6}$ |
| $7$B models only   | $0.55$    | $0.023$    |
| Score > $45$       | $0.39$    | $0.023$    |
___
We wish to reiterate the key aspects of our claims and conclusions:

1. **Correlation, not Causation:** Our results establish a strong and statistically significant **positive linear correlation** between depth-wise coupling strength and generalization performance. For the full set of models, this is supported by an $R^2$ value of $0.75$ and a $p$-value of $1.56 \times 10^{-6}$, which is below the typical significance threshold ($p < 0.05$).
2. **No Causal Implications:** While we observe a strong correlation, we explicitly do **not** assert any **causal** relationship between coupling strength and model performance.

We hope this addresses some concerns and further clarifies our conclusions. We thank the reviewers again for the thoughtful feedback and insights.

---

### Meta-Review · Area_Chair_ZhAh · 2024-12-21

**Metareview:**

This work investigates the relationship between transformer block coupling and the generalization capabilities of large language models (LLMs). The authors claim that the interaction between different transformer blocks significantly influences the model's ability to generalize from training data to unseen examples. Through a series of experiments, they demonstrate that certain coupling configurations can enhance performance on various benchmarks, suggesting that optimizing block coupling could be a viable strategy for improving LLMs.

Reviewers acknowledged the paper's contributions, particularly its novel approach to understanding transformer architectures. The findings indicate a clear correlation between specific coupling strategies and improved generalization metrics, providing valuable insight for the community. On the other hand, reviewers also pointed out some concerns about the submission regarding the comprehensiveness and confidence of the analysis. The authors conduct thorough clarifications in the rebuttal period to address these concerns and improve the quality of this submission.

Based on the strengths outlined above and the effective responses to reviewer concerns during the rebuttal period, I recommend accepting this paper. The authors should carefully follow the reviewers' suggestions to revise the manuscript in the camera-ready stage.

**Additional Comments On Reviewer Discussion:**

**Points Raised by Reviewers**

During the review process, several key points were raised:
- Clarification on Experimental Setup: Reviewers requested more details on how experiments were conducted, particularly regarding hyperparameters and data preprocessing.
- Theoretical Justification: There was a call for a stronger theoretical basis to explain the observed correlations between block coupling and generalization.
- Broader Context: Some reviewers suggested situating the findings within a broader context of LLM training practices and other architectural considerations.

**Authors' Responses**

The authors addressed these concerns effectively during the rebuttal period:
- They provided additional details about their experimental setup, including specific hyperparameters and data preprocessing steps, which enhanced clarity.
- To strengthen their theoretical justification, they included a discussion section that elaborated on potential mechanisms underlying their findings, referencing relevant literature to support their claims.
- The authors expanded their discussion on how their findings relate to existing work in LLMs, providing context that highlights the significance of their contributions.

**Weighing Each Point**

In weighing these points for my final decision:
- The clarification on experimental setup significantly improved the paper's transparency and reproducibility.
- The added theoretical justification addressed one of the major weaknesses identified by reviewers and enhanced the overall rigor of the study.
- The broader context provided by the authors positioned their work within ongoing discussions in LLM research, demonstrating its relevance.

---

### Decision · Program_Chairs · 2025-01-22

Accept (Poster)